# Distributed Epigraph Form Multi-Agent Safe Reinforcement Learning

## Abstract

Most existing safe multi-agent reinforcement learning (MARL) algorithms consider the constrained Markov decision process (CMDP) problem, which targets bringing the mean of constraint violation below a user-defined threshold. However, as observed by existing works albeit for the single-agent case, CMDP algorithms suffer from unstable training when the constraint threshold is zero. This paper proposes **EFMARL**, a novel MARL algorithm that improves upon the problems faced in the zero constraint threshold setting by extending the *epigraph form*, a technique to perform constrained optimization, to the centralized training and distributed execution (CTDE) paradigm. We validate our approach in different Multi-Particle Environments and Safe Multi-agent MuJoCo environments with varying numbers of agents. Simulation results show that our algorithm achieves stable training and the best performance while satisfying constraints: it is as safe as the safest baseline that has significant performance loss, and achieves similar performance as baselines that prioritize performance but violate safety constraints.

## 1 Introduction

Multi-agent systems (MAS) play an integral role in our aspirations for a more convenient future with examples such as autonomous warehouse operations (Kattepur et al., 2018), large-scale autonomous package delivery (Ma et al., 2017), traffic routing (Wu et al., 2020), and power systems (Biagioni et al., 2022). For MAS, distributed policies are desirable due to their ability to scale to a large number of agents compared to centralized policies (Pereira et al., 2022; Saravanos et al., 2023). To construct such policies, multi-agent reinforcement learning (MARL) (Zhang et al., 2021; Garg et al., 2024) has emerged as an attractive method. While the learned policies must be safe for real-world deployment, many MARL algorithms do not explicitly consider safety constraints (Sunehag et al., 2017; Rashid et al., 2020b; Yang et al., 2020; Wang et al., 2020; Peng et al., 2021; Rashid et al., 2020a), but instead optimizing for a single objective that must be designed to incorporate safety. Although safe MARL methods have been developed in recent years (Gu et al., 2023; Liu et al., 2021; Ding et al., 2023; Lu et al., 2021; Geng et al., 2023; Zhao et al., 2024), most of these methods target the constrained Markov decision process (CMDP) (Altman, 2004) setting, which only asks for the mean constraint violation to stay below a user-defined threshold. This is unacceptable for safety-critical applications such as autonomous vehicles or human-robot interactions, where any constraint violation can be fatal. While this can be addressed by setting the constraint violation threshold to *zero* in the CMDP, in this setting the popular Lagrangian methods experience training instabilities which result in sharp drops in performance during training, and non-convergence or convergence to poor policies (So & Fan, 2023; He et al., 2023; Ganai et al., 2024; Huang et al., 2024).

These concerns have been identified recently, resulting in a series of works that enforce hard constraints (Zanon & Gros, 2020; Zhao et al., 2021; So & Fan, 2023; Ganai et al., 2024) using techniques inspired by Hamilton-Jacobi reachability (Tomlin et al., 2000; Lygeros, 2004; Mitchell et al., 2005; Margellos & Lygeros, 2011; Bansal et al., 2017) in deep reinforcement learning (RL) for the *single-agent* case and have been shown to significantly improve safety compared to other safe RL approaches. However, to the best of our knowledge, theories, and algorithms for safe RL are still lacking for the *multi-agent* scenario, especially when policies are executed in a distributed manner. While single-agent RL methods can be directly applied to the MARL setting by treating the MAS as a centralized single agent, the joint action space grows exponentially with the number of agents,

preventing these algorithms from scaling to scenarios with a large number of agents (Guestrin et al., 2002; Sunehag et al., 2017; Foerster et al., 2018).

To tackle the problem of zero constraint violation in multi-agent scenarios with *distributed* policies[1] while achieving high collaborative performance, we propose **Epigraph Form MARL (EFMARL)**. EFMARL directly tackles the multi-agent constrained optimal control problem (MACOCP), whose solution satisfies zero constraint violation. To solve the MACOCP, EFMARL uses the epigraph form technique (Boyd & Vandenberghe, 2004), which has previously been shown to result in better policies compared to Lagrangian-based methods (So & Fan, 2023) in the single-agent setting. Considering the multi-agent setting, we propose an extension of the epigraph form that falls under the centralized training distributed execution (CTDE) paradigm.

We validate EFMARL using various tasks from multi-particle environments (MPE) (Lowe et al., 2017) and Safe Multi-agent MuJoCo (Gu et al., 2023) with varying numbers of agents and compare its performance with existing safe MARL algorithms using the penalty and Lagrangian methods. The results suggest that EFMARL achieves the best performance while satisfying constraints: it is as safe as conservative baselines that achieve high safety but sacrifice performance, while matching the performance of unsafe baselines that sacrifice safety for high performance. In addition, while the baseline methods require different choices of hyperparameters to perform well in each environment and suffer from unstable training because of zero constraint violation threshold, EFMARL is stable in training using the same hyperparameters across all environments, indicative of the algorithm's robustness to environmental changes. To summarize, our contributions are presented below:

- Drawing on prior work that addresses the training instability of Lagrangian methods in the zero-constraint violation setting, we extend the epigraph form method from single-agent RL to MARL, improving upon the training instability of existing MARL algorithms.

- We present theoretical results showing that the outer problem of the epigraph form can be decomposed and solved in a distributed manner during online execution. This allows EFMARL to fall under the CTDE paradigm.

- We illustrate through extensive simulations that, without any hyperparameter tuning, EFMARL achieves stable training and is as safe as the most conservative baseline while simultaneously being as performant as the most aggressive baseline across all environments.

## 2 RELATED WORK

**Shielding for Safe MARL** One popular method of providing safety to learning-based methods is using *shielding* or a *safety filter* (Garg et al., 2024). Here, an unconstrained learning method is paired with a shield or safety filter using techniques such as predictive safety filters (Zhang et al., 2019; Muntwiler et al., 2020), control barrier functions (Cai et al., 2021; Pereira et al., 2022) or automata (ElSayed-Aly et al., 2021a; Xiao et al., 2023; Melcer et al., 2022). Such shields are often constructed before learning begins and are used to modify either the feasible actions or the output of the learned policy to maintain safety. One benefit is that safety can be guaranteed during both training and deployment since the shield is constructed before training. However, they require domain expertise to build a valid shield, which can be challenging in the single-agent setting and even more difficult for MAS (Garg et al., 2024). Other methods can automatically synthesize shields but face scalability challenges (Melcer et al., 2022; ElSayed-Aly et al., 2021b). Another drawback is that the policy after shielding might not consider the same objective as the original policy and may result in noncollaborative behaviors or deadlocks (Qin et al., 2021; Zhang et al., 2023; 2024).

**Unconstrained MARL** Early works that approached the problem of safety for MARL focused on navigation problems and collision avoidance (Chen et al., 2017b;a; Everett et al., 2018; Semnani et al., 2020), where safety is achieved by a sparse collision penalty (Long et al., 2018), or a shaped reward penalizing getting close to obstacles and neighboring agents (Chen et al., 2017b;a; Everett et al., 2018; Semnani et al., 2020). However, adding a penalty to the reward function changes the original objective, so the resulting policy may not be optimal for the original constraint optimization

---

[1] In this paper, the policies are distributed if each agent makes decisions using local information/sensor data and information received via message passing with other agents (Garg et al., 2024), although this setting is sometimes called "decentralized" in MARL (Zhang et al., 2018).

problem. In addition, the satisfaction of collision avoidance constraints is not necessarily guaranteed by even the optimal policy (Massiani et al., 2023; Everett et al., 2018; Long et al., 2018).

**Constrained MARL** In contrast to unconstrained MARL methods, which change the constraint optimization problem to an unconstrained problem, constrained MARL methods explicitly solve the CMDP problem. For the single-agent case, prominent methods for solving CMDPs include primal methods (Xu et al., 2021), primal-dual methods using Lagrange multipliers (Borkar, 2005; Tessler et al., 2019; He et al., 2023; Huang et al., 2024), and via trust-region-based approaches (Achiam et al., 2017; He et al., 2023). These methods provide guarantees either in the form of asymptotic convergence guarantees to the optimal (safe) solution (Borkar, 2005; Tessler et al., 2019) using stochastic approximation theory (Robbins & Monro, 1951; Borkar, 2009), or recursive feasibility of intermediate policies (Achiam et al., 2017; Satija et al., 2020) using ideas from trust region optimization (Schulman et al., 2015a). The survey (Gu et al., 2022) provides an overview of the different methods of solving safety-constrained single-agent RL. In multi-agent cases, however, the problem becomes more difficult because of the non-stationary behavior of other agents, and similar approaches have been presented only recently (Gu et al., 2023; Liu et al., 2021; Ding et al., 2023; Lu et al., 2021; Geng et al., 2023; Zhao et al., 2024; Chen et al., 2024). However, the CMDP setting they handle makes it difficult for them to handle hard constraints, and results in poor performance when the constraint violation threshold is zero (Ganai et al., 2024).

## 3 PROBLEM SETTING AND PRELIMINARIES

### 3.1 MULTI-AGENT CONSTRAINED OPTIMAL CONTROL PROBLEM

We consider the multi-agent constrained optimal control problem defined as follows. Consider a homogeneous MAS with $N$ agents. At time step $k$, the global state and control input is given by $x^k \in \mathcal{X} \subseteq \mathbb{R}^n$ and $u^k \in \mathcal{U} \subseteq \mathbb{R}^m$. The global control vector is defined by concatenation $u^k := [u_1^k; \ldots; u_N^k]$, where $u_i^k \in \mathcal{U}_i$ is the control input of agent $i$. We consider nonlinear discrete-time dynamics for the MAS:

$$x^{k+1} = f(x^k, u^k), \tag{1}$$

where $f : \mathcal{X} \times \mathcal{U} \to \mathcal{X}$ is the global dynamics function. We consider the partially observable setting, where each agent has a limited communication radius $R > 0$ and can only communicate with other agents or observe the environment within its communication region. Denote $o_i^k = O_i(x^k) \in \mathcal{O} \subseteq \mathbb{R}^{n_o}$ as the vector of the information observed by agent $i$ at the time step $k$, where $O_i : \mathcal{X} \to \mathcal{O}$ is an encoding function of the information shared from neighbors of agent $i$ and the observed data of the environment. We allow multi-hop communication between agents, so an agent can communicate with another agent outside its sensing region if a communication path exists between them.

Let the avoid/unsafe set of agent $i$ be $\mathcal{A}_i := \{o_i \in \mathcal{O} : h_i(o_i) > 0\}$, for some function $h_i : \mathcal{O} \to \mathbb{R}$. The global avoid set is then defined as $\mathcal{A} := \{x \in \mathcal{X} : h(x) > 0\}$, where $h(x) = \max_i h_i(o_i) = \max_i h_i(O_i(x))$. In other words, $\exists i$, s.t. $o_i \in \mathcal{A}_i \iff x \in \mathcal{A}$. Given a global cost function $l : \mathcal{X} \times \mathcal{U} \to \mathbb{R}$ describing the task for the agents to accomplish [2], we aim to find decentralized control policies $\pi_i : \mathcal{O} \to \mathcal{U}_i$ such that starting from any given initial states $x^0 \notin \mathcal{A}$, the policies keep the agents outside the avoid set $\mathcal{A}$ and minimize the infinite horizon cost $\sum_{k=0}^{\infty} l(x^k, \pi(x^k))$. In other words, denoting $\pi : \mathcal{X} \to \mathcal{U}$ as the joint policy such that $\pi(x) = [\pi_1(o_1); \ldots; \pi_N(o_N)] = [\pi_1(O_1(x)); \ldots; \pi_N(O_N(x))]$, we aim to solve the following infinite-horizon multi-agent constrained optimal control problem (MACOCP) for a given initial state $x^0$:

$$\min_{\{\pi_i\}_{i=1}^N} \quad \sum_{k=0}^{\infty} l(x^k, \pi(x^k)) \tag{2a}$$

$$\text{s.t.} \quad h_i(o_i^k) \leq 0, \quad o_i^k = O_i(x^k), \qquad \forall i \in \{1, \ldots, N\}, k \geq 0, \tag{2b}$$

$$x^{k+1} = f(x^k, \pi(x^k)), \qquad k \geq 0. \tag{2c}$$

Note that the safety constraint (2b) differs from the average constraints considered in CMDPs (Altman, 2004). Consequently, instead of allowing safety violations to occur as long as the mean con-

---

[2] The cost function $l$ is **not** the cost in CMDP. Rather, it corresponds to the negation of the *reward* in CMDP.

straint violation is below a threshold, this formulation disallows *any* constraint violation. From hereon after, we omit the dynamics constraint (2c) for conciseness.

## 3.2 EPIGRAPH FORM

Existing methods are unable to solve (2) well. This has been observed previously in the single-agent setting (Zanon & Gros, 2020; Zhao et al., 2021; So & Fan, 2023; Ganai et al., 2024). We show later that the poor performance of methods that tackle the CMDP setting to the constrained problem (2) also translates to the multi-agent setting, as we observe a similar phenomenon in our experiments (Section 5). Namely, although unconstrained MARL can be used to solve (2) using the penalty method (Nayak et al., 2023), this does not perform well in practice, where a small penalty results in policies that violate constraints, and a large penalty results in higher total costs. The Lagrangian method (Gu et al., 2023) can solve the problem theoretically, but it suffers from unstable training and has poor performance in practice when the constraint violation threshold is zero (So & Fan, 2023; Ganai et al., 2024). In this section, we introduce a new method of solving (2) that can mitigate the above problems by extending prior work (So & Fan, 2023) to the multi-agent setting.

Given a constrained optimization problem with objective function $J$ (e.g., $J = \sum_{k=0}^{\infty} l$ as in (2a)), and constraints $h$ (e.g., (2b)):

$$\min_{\pi} \quad J(\pi) \qquad \text{s.t.} \quad h(\pi) \leq 0, \tag{3}$$

its epigraph form (Boyd & Vandenberghe, 2004) is given as

$$\min_{\pi, z} \quad z \qquad \text{s.t.} \quad h(\pi) \leq 0, \quad J(\pi) \leq z, \tag{4}$$

where $z \in \mathbb{R}$ is an auxiliary variable. In other words, we add a constraint to enforce $z$ as an upper bound of the cost $J(\pi)$, then minimize the upper bound $z$. The solution to (4) is identical to the original (3) (Boyd & Vandenberghe, 2004). Furthermore, (4) is equivalent (So & Fan, 2023) to

$$\min_{z} \quad z \tag{5a}$$

$$\text{s.t.} \quad \min_{\pi} J_z(\pi, z) \leq 0, \qquad J_z(\pi, z) := \max\{h(\pi), J(\pi) - z\} \tag{5b}$$

As a result, the original constrained problem (3) is decomposed into the following two subproblems:

1. An unconstrained *inner problem* (5b), where, given an arbitrary desired cost upper bound $z$, we find $\pi$ such that $J_z(\pi, z)$ is minimized, i.e., best satisfies the constraints $h \leq 0$ and $J \leq z$.

2. A 1-dimensional constrained *outer problem* (5a) over $z$ to find the smallest cost upper bound $z$ such that $z$ is a cost upper bound ($J \leq z$) and the constraints $h(\pi) \leq 0$ holds.

**Comparison with the Lagrangian method.** Another popular way to solve MACOCP (2) is the Lagrangian method (Gu et al., 2023). However, it suffers from unstable training when considering the *zero* constraint violation (So & Fan, 2023; He et al., 2023) setting. More specifically, this refers to the case with constraints $\sum_{k=0}^{\infty} c(x^k) \leq 0$ for $c : \mathcal{X} \to \mathbb{R}_{\geq 0}$ non-negative. Since $h$ can be negative, we can convert our problem setting (3) to the zero constraint violation setting by taking $c(x) := \max\{0, h(x)\}$. Then, (3) reads as

$$\min_{\pi} \quad J(\pi) \qquad \text{s.t.} \quad \sum_{k=0}^{\infty} \max\{0, h(x^k)\} \leq 0. \tag{6}$$

The Lagrangian form of (6) is then

$$\max_{\lambda \geq 0} \min_{\pi} \quad J_\lambda(\pi, \lambda) := J(\pi) + \lambda \sum_{k=0}^{\infty} \max\{h(x^k), 0\}, \tag{7}$$

where $\lambda$ is the Lagrangian multiplier and is updated with gradient ascent. However, $\frac{\partial}{\partial \lambda} J_\lambda(\pi, \lambda) = \sum_{k=0}^{\infty} \max\{h(x^k), 0\} \geq 0$, so $\lambda$ continuously increases and never decreases. As $\frac{\partial}{\partial \pi} J_\lambda(\pi, \lambda)$ scales linearly in $\lambda$ when $h(x^k) > 0$ for some $k$, a large value of $\lambda$ causes a large gradient w.r.t $x$, and makes the training unstable. Note that for the epigraph form, since $z$ does not *multiply* with the cost function $J$ but is *added* to $J$ in (5b), the gradient $\frac{\partial}{\partial \pi} J_z(\pi, z)$ does not scale with the value of $z$ resulting in more stable training. We validate this in our experiments (Section 5).

## 4 EPIGRAPH FORM MULTI-AGENT REINFORCEMENT LEARNING

In this section, we propose the Epigraph Form MARL (EFMARL) algorithm to solve MACOCP (2) using MARL. First, we transfer MACOCP (2) to its epigraph form with an auxiliary variable $z$ to model the desired cost upper bound. To fit the CTDE paradigm, we propose a centralized inner problem to jointly train the agents' policies given the desired cost upper bound $z$, and a distributed outer problem in execution to find the smallest cost upper bound $z$ that ensures safety.

### 4.1 EPIGRAPH FORM FOR MACOCP

To rewrite MACOCP (2) into its epigraph form (5), we first define the cost-value function $V^l$ for a joint policy $\pi$ using the standard optimal control notation (Bertsekas, 2012):

$$V^l(x^\tau; \pi) := \sum_{k \geq \tau} l(x^k, \pi(x^k)). \tag{8}$$

We also define the constraint-value function $V^h$ as the maximum constraint violation:

$$V^h(x^\tau; \pi) := \max_{k \geq \tau} h(x^k) = \max_{k \geq \tau} \max_i h_i(o_i^k) = \max_i \max_{k \geq \tau} h_i(o_i^k) = \max_i V_i^h(o_i^\tau; \pi). \tag{9}$$

Here, we interchange the $\max$ to define the *local per-agent* functions $V_i^h(o_i^\tau; \pi) = \max_{k \geq \tau} h_i(o_i^k)$. Each $V_i^h$ uses only the agent's local observation and thus is distributed. We now introduce the auxiliary variable $z$ for the desired upper bound of $V^l$, allowing us to restate (2) concisely as

$$\min_{\{\pi_i\}_{i=1}^N} V^l(x^0; \pi) \quad \text{s.t.} \quad V^h(x^0; \pi) \leq 0. \tag{10a}$$

The epigraph form (5) of (10) then takes the form

$$\min_z \; z \quad \text{(11a)} \qquad \text{s.t.} \quad \min_{\{\pi_i\}_{i=1}^N} \underbrace{\max\left\{\max_i V_i^h(o_i^\tau; \pi), V^l(x^\tau; \pi) - z\right\}}_{:=V(x^0, z; \pi)} \leq 0. \tag{11b}$$

By interpreting the left-hand side of (11b) as a *new* policy optimization problem, we define the *total value function* $V$ as the objective function to (11b). This can be simplified as

$$V(x^\tau, z; \pi) = \max\{\max_i V_i^h(o_i^\tau; \pi), V^l(x^\tau; \pi) - z\} \tag{12a}$$

$$= \max_i \max\{V_i^h(o_i^\tau; \pi), V^l(x^\tau; \pi) - z\} = \max_i V_i(x^\tau, z; \pi), \tag{12b}$$

Again, we interchange the $\max$ to define $V_i(x^\tau, z; \pi) = \max\{V_i^h(o_i^\tau; \pi), V^l(x^\tau; \pi) - z\}$ as the *per-agent* total value function. Using this to rewrite (11) then yields

$$\min_z \; z \quad \text{(13a)} \qquad\qquad \text{s.t.} \quad \min_\pi \max_i V_i(x^0, z; \pi) \leq 0. \tag{13b}$$

This decomposes the original problem (2) into an unconstrained inner problem (13b) over policy $\pi$ and a constrained outer problem over $z$. During offline training, we solve the inner problem (13b): for parameter $z$, find the optimal policy $\pi(\cdot, z)$ to minimize $V(x^0, z; \pi)$. Note that the optimal policy of the inner problem depends on $z$. During execution, we solve the outer problem (13a) online to get the minimal $z$ that satisfies constraint (13b). Using this $z$ in the $z$-conditioned policy $\pi(\cdot, z)$ found in the inner problem gives us the optimal policy for the overall EF-MACOCP.

To solve the inner problem (13b), the total value function $V$ must be amenable to dynamic programming, which we show in the following proposition.

**Proposition 1.** *Dynamic programming can be applied to EF-MACOCP (13), resulting in*

$$V(x^k, z^k; \pi) = \min_{u^k} \max\{h(x^k), V(x^{k+1}, z^{k+1}; \pi)\}, \quad z^{k+1} = z^k - l(x^k, \pi(x^k)). \tag{14}$$

The proof of Proposition 1 is provided in Appendix A. In other words, for a given cost upper bound $z^k$, the value function $V$ at the current state $x^k$ can be computed using the value function at the next state $x^{k+1}$ but with a *different* cost upper bound $z^{k+1} = z^k - l(x^k, \pi(x^k))$ which itself is a function of $z^k$. This can be interpreted as a "*dynamics*" for the cost upper bound $z$. Intuitively, if we wish to satisfy the upper bound $z^k$ but suffer a cost $l(x^k, \pi(x^k))$, then the upper bound at the next time step should be smaller by $l(x^k, \pi(x^k))$ so that the total cost from $x^k$ remains upper bounded by $z^k$. Additional discussion on Proposition 1 is provided in Appendix C.

Figure 1: **EFMARL algorithm.** Randomly sampled initial states and $z^0$ are used to collect trajectories in $x$ and $z$ using the current policy $\pi$. In the centralized training (orange blocks), distributed constraint-value functions $V_i^h$ and policies $\pi_i$ and a centralized cost-value function $V^l$ are jointly trained. During distributed execution (green blocks), the distributed $V_i^h$ are used to solve the outer problem (15b) to compute the optimal $z_i$, which is used in each agent's $z$-conditioned policy.

**Remark 1** (Effect of $z$ on the learned policy). *From (12), for a fixed $x$ and $\pi$, observe that for $z$ large enough (i.e., $V^l(x;\pi) - z$ is small enough), then $V(x, z; \pi) = V^h(x; \pi)$. Consequently, taking a gradient step on $V(x, z; \pi)$ equals to gradient steps on $V^h(x; \pi)$, reducing the constraint violation possibly in exchange for an increase in the total cost $V^l(x; \pi)$. Otherwise, $V(x, z; \pi) = V^l(x; \pi)$. Taking gradient steps on $V(x, z; \pi)$ equals to gradient steps on $V^l(x; \pi)$, reducing the total cost possibly in exchange for larger constraint violation.*

### 4.2 SOLVING THE INNER PROBLEM USING MARL

Following So & Fan (2023), we solve the inner problem using centralized training with proximal policy optimization (PPO) (Schulman et al., 2017). We use a graph neural network (GNN) backbone for the $z$-conditioned policy $\pi_\theta(o_i, z)$, cost-value function $V_\phi^l(x, z)$, and the constraint-value function $V_\psi^h(o_i, z)$ with parameters $\theta$, $\phi$, and $\psi$, respectively. Note that other NN structures can be used as well. The implementation details are introduced in Appendix E.

**Policy and value function updates** During centralized training, the NNs are trained to solve the inner problem (13b), i.e., for a randomly sampled $z$, find policy $\pi(\cdot, z)$ that minimizes the total value function $V(x^0, z; \pi)$. We follow MAPPO (Yu et al., 2022a) to train the NNs ~~with advantage decomposition (Gu et al., 2023)~~. Specifically, when calculating the advantage for $i$-th agent, $A_i$ (Schulman et al., 2017), with the generated advantage estimation (GAE) (Schulman et al., 2015b) for each agent, instead of using the cost function $V^l$ (Yu et al., 2022a), we apply the decomposed total value function $\max\{V_\psi^h(o_i, z), V_\phi^l(x, z) - z\}$. We perform trajectory rollouts following the dynamics for $x$ (1) and $z$ (14) using the learned policy $\pi_\theta$, starting from random sampled $x^0$ and $z^0$. After collecting the trajectories, we train the cost-value function $V_\phi^l$ and the constraint-value function $V_\psi^h$ via regression and use the PPO policy loss to update the $z$-conditioned policy $\pi_\theta$.

### 4.3 SOLVING THE OUTER PROBLEM DURING DISTRIBUTED EXECUTION

During execution, we solve the outer problem of EF-MACOCP (13) online. However, the outer problem is still centralized because the constraint (13b) requires the centralized cost-value function $V^l$. To achieve a distributed policy during execution, we introduce the following theoretical result:

**Theorem 1.** *The outer problem of EF-MACOCP (5a) is equivalent to the following problem:*

$$z = \max_i z_i \tag{15a}$$

$$\text{s.t.} \quad z_i = \arg\min z' \quad \text{s.t.} \quad V_i^h(o_i; \pi(\cdot, z')) \leq 0, \quad i = 1, \cdots, N. \tag{15b}$$

The proof is provided in Appendix B. Theorem 1 enables computing $z$ without the use of the centralized $V^l$ during execution. Specifically, each agent $i$ solves the local problem (15b) for $z_i$, which

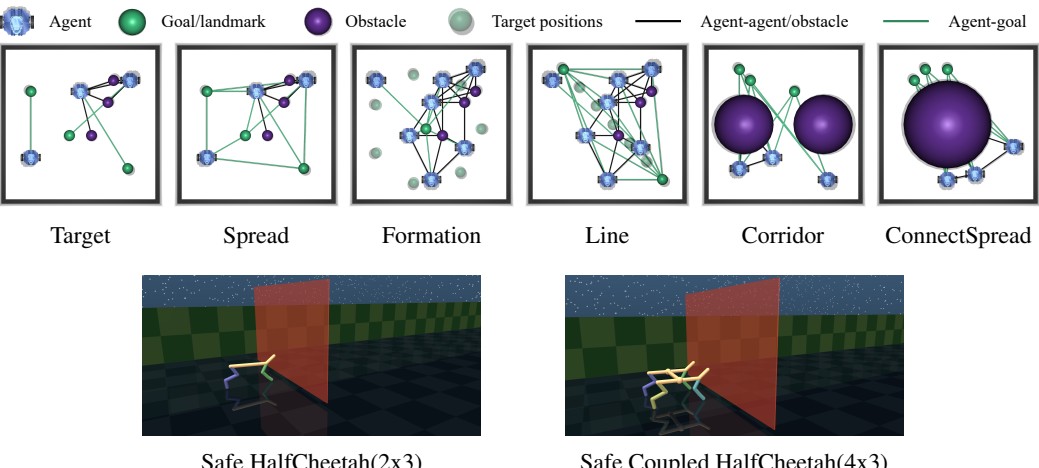

Figure 2: **Simulation Environments.** Visualizations of the (top) *modified* MPE Lowe et al. (2017) and (bottom) Safe Multi-agent MuJoCo Gu et al. (2023) environments we consider.

is a 1D optimization problem and can be efficiently solved using root-finding methods (e.g., Chandrupatla (1997)) as in So & Fan (2023), then communicates $z_i$ among the other agents to obtain the maximum (15a). One challenge is that this maximum may not be computable if the agents are not connected. However, in our problem setting, if one agent is not connected, it does not appear in the observations $o$ of the connected agent. Therefore, it would not contribute to the $V^h$ of other agents. As a result, it is sufficient for only the connected agents to communicate their $z_i$. Furthermore, we observe experimentally that the agents can achieve low cost while maintaining safety even if $z_i$ is not communicated (see Section 5.3). Thus, we do not include $z_i$ communication for our method.

Since there may be errors estimating $V^h$ using NN, we can reduce the resulting safety violation by modifying $h$ to add a buffer region. Specifically, for a constant $\nu > 0$, we modify $h$ such that $h \geq \nu$ when the constraints are violated and $h \leq -\nu$ otherwise. We then modify (15b) to $V^h_\psi(o_i, z_i) \leq -\xi$, where $\xi \in [0, \nu]$ is a hyperparameter (where we want $\xi \approx \nu$ to emphasize more on safety). This makes $z$ more robust to estimation errors of $V^h$. We study the importance of $\xi$ in Section 5.3.

## 5 EXPERIMENTS

In this section, we design experiments to answer the following research questions:

1. Does EFMARL satisfy safety constraints and achieve low cost with constant hyperparameters across all environments?
2. How does EFMARL behave compared with baselines with different hyperparameters considering performance, safety, and training stability?
3. Does EFMARL maintain high performance and safety with an increasing number of agents?

Details for the implementation, environments, and hyperparameters are provided in Appendix E.

### 5.1 SETUP

We evaluate EFMARL in two sets of environments: a *modified* MPE (Lowe et al., 2017), and Safe Multi-agent MuJoCo (Gu et al., 2023) (see Figure 2). In MPE, the agents are assumed to have double integrator dynamics with bounded *continuous* action spaces $[-1, 1]^2$. We provide the full details of all tasks in Appendix E. To increase the difficulty of the tasks, we add 3 static obstacles to these environments. For Safe Multi-agent MuJoCo environments, we consider **HalfCheetah 2x3** and **Coupled HalfCheetah 4x3**. The agents must collaborate to make the cheetah run as fast as possible without colliding with a moving wall in front. To design the constraint function $h$, we let $\nu = 0.5$ in all our experiments and $\xi = 0.4$ when solving the outer problem.

**Baselines** We compare our algorithm with the state-of-the-art (SOTA) MARL algorithm Infor-MARL (Nayak et al., 2023) with a constraint-penalized cost $l'(x, u) = l(x, u) + \beta \max\{h(x), 0\}$,

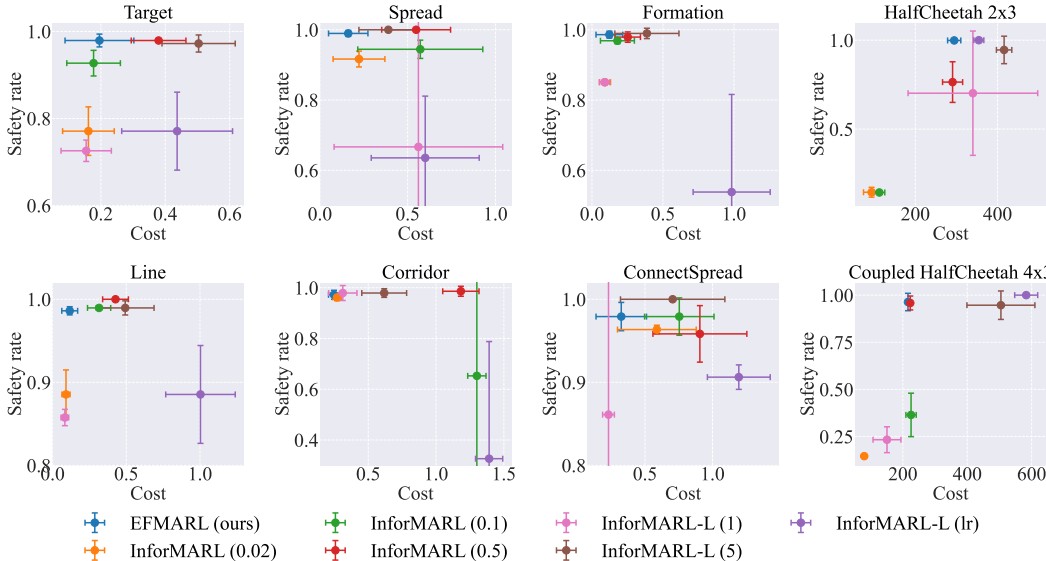

Figure 3: **Comparison on modified MPE ($N = 3$) and Safe Multi-agent MuJoCo.** EFMARL is consistently closest to the top-left corner in all environments, achieving low cost with near $100\%$ safety rate. The dots show the mean values and the error bar shows one standard deviation.

where $\beta \in \{0.02, 0.1, 0.5\}$ is a penalty parameter, and denote this baseline as InforMARL ($\beta$). We also consider the SOTA safe MARL algorithm MAPPO-Lagrangian (Gu et al., 2021; 2023)[3]. In addition, because the learning rate of the Lagrangian multiplier $\lambda$ is tiny ($10^{-7}$) in the official implementation[4] of MAPPO-Lagrangian (Gu et al., 2023), the value of $\lambda$ during training will be largely determined by the initial value $\lambda_0$ of $\lambda$. We thus consider two $\lambda_0 \in \{1, 5\}$. Moreover, to compare the training stability, we consider increasing the learning rate of $\lambda$ in MAPPO-Lagrangian to $3 \times 10^{-3}$.[5] For a fair comparison, we reimplement MAPPO-Lagrangian using the same GNN backbone as used in EFMARL and InforMARL, denoted as InforMARL-L ($\lambda_0$) and InforMARL-L (lr) for the increased learning rate one. All algorithms use an RNN for the final layer.

**Evaluation criteria** Following the goal of the MACOCP, we use the cost and safety rate as the evaluation criteria for the performances of all algorithms. The **cost** is the cumulative cost over the trajectory $\sum_{k=0}^{T} l(x^k, u^k)$. The **safety rate** is defined as the ratio of agents that remain safe over the entire trajectory, i.e., $h_i(o_i^k) \leq 0, \forall k$, over all agents. Unlike the CMDP setting, we do not report the mean of constraint violations over time but the violation of the hard safety constraints.

## 5.2 RESULTS

We train all algorithms with 3 different random seeds and test the converged policies on 32 different initial conditions. As discussed in Section 4.3, we disable the communication of $z_i$ between agents (investigated in Section 5.3). The safety rate (y-axis) and cumulative cost (x-axis) for each algorithm are plotted in Figure 3. Thus, the closer an algorithm is to the top-left corner, the better it performs. In both MPE and safe Multi-agent MuJoCo environments, EFMARL is always closest to the top-left corner, maintaining a low cost while having near $100\%$ safety rate. While InforMARL with $\beta = 0.02$ and InforMARL with $\lambda_0 = 1$ generally have low costs, they also have frequent constraint violations. With $\beta = 0.5$ or $\lambda_0 = 5$, they prioritize safety but at the cost of high cumulative costs. EFMARL, however, maintains a safety rate similar to the most conservative baselines (InforMARL (0.5) and InforMARL-L (5)) but has much lower costs. We point out that no *single* baseline method behaves considerably better on *all* the environments: the performance of the baseline methods varies wildly between environments, demonstrating the sensitivity of these algorithms to the choice of

---

[3]We omit the comparison with MACPO (Gu et al., 2021; 2023) as it was shown to perform similarly to MAPPO-Lagrangian but have significantly worse time complexity and wall clock time for training.

[4]https://github.com/chauncygu/Multi-Agent-Constrained-Policy-Optimisation

[5]This is the smallest learning rate for $\lambda$ that does not make MAPPO-Lagrangian ignore the safety constraint.

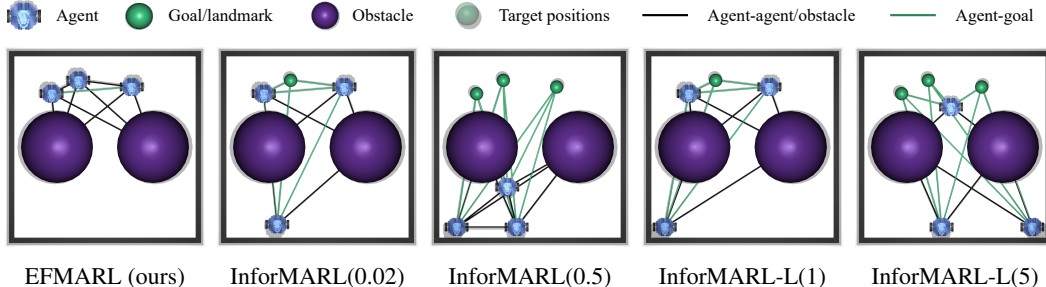

Figure 4: **Converged states in Corridor.** EFMARL achieves the global minimum, while other baselines converge to a different optimum (partly) due to training using a *different* cost function.

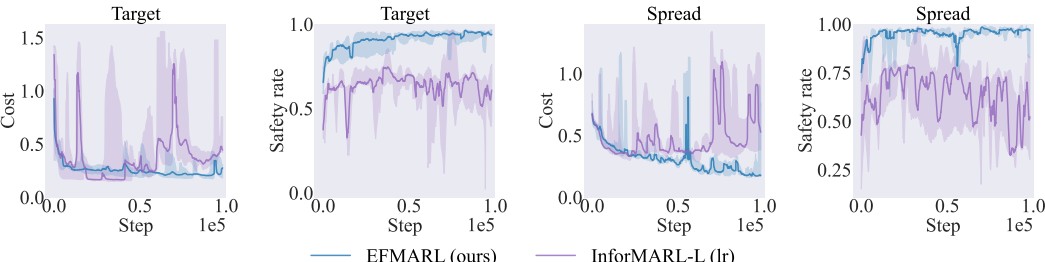

Figure 5: **Training Curves in Target and Spread.** EFMARL has a smoother, more stable training curve compared to InforMARL-L (lr). We plot the mean and shade the $\pm 1$ standard deviation.

hyperparameters. EFMARL, using a single set of constant hyperparameters, performs best in *all* environments, demonstrating its insensitivity to the choice of hyperparameters.

An important observation is that for InforMARL ($\beta$) and InforMARL-L with a non-optimal $\lambda$, the cost function optimized in their training process is *different* from the original cost function. Consequently, they can have different optimal solutions compared to the original problem. Therefore, even if their training converges, they may not reach the optimal solution of the original problem. In Figure 4, the converged states of EFMARL and four baselines are shown. EFMARL reaches the original problem's global optimum and covers all three goals. On the contrary, the optimums of InforMARL (0.02) and InforMARL-L (1) are *changed* by the penalty term, so they choose to leave one agent behind to have a lower safety penalty. With an even more significant penalty, the optimums of InforMARL (0.5) and InforMARL-L (5) are changed dramatically, and they forget the goal entirely and only focus on safety.

To compare the training stability of EFMARL and InforMARL-L, we plot the cost and safety rate of EFMARL and InforMARL-L (lr) during training in Figure 5. EFMARL has a *smoother* curve compared to InforMARL-L (lr), supporting our theoretical analysis in Section 3.2. Due to space limits, the plots for other environments and other baseline methods are provided in Appendix E.5.

To evaluate the performance of EFMARL in environments with more agents, we also train the algorithms with 5 and 7 agents in the Formation and the Line environments. The results are shown in Figure 6. Because InforMARL-L (lr) has the worst performance in MPE with $N = 3$, we omit it in this experiment. EFMARL is closest to the upper left corner in all environments, and its performance does not decrease with an increasing number of agents.

### 5.3 ABLATION STUDIES

**Is communicating $z_i$ necessary?** As introduced in Section 4.3, theoretically, all connected agents should communicate and reach a consensus on $z = \max_i z_i$. However, we observe in Section 5.2 that the agents can perform well even if agents take $z \leftarrow z_i$ without communicating to compute the maximum. We perform experiments on Line ($N = 3$) to understand the impact of this approximation in Table 1 and see that using the approximation does not result in much performance difference compared to communicating $z_i$ and using the maximum.

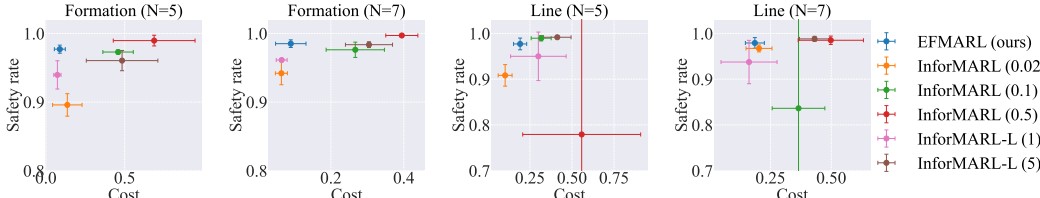

Figure 6: **Comparison on modified MPE ($N = 5, N = 7$).** EFMARL remains in the top-left corner even when the number of agents is increased with similar performance across both $N = 5$ and $N = 7$. The dots show the mean and the error bar shows one standard deviation.

Table 1: Effect of $z_i$ communication (Section 4.3) in different environments.

| Environment | No communication ($z \leftarrow z_i$) | | Communication ($z = \max_i z_i$) | |
|---|---|---|---|---|
| | Safety rate | Cost | Safety rate | Cost |
| Target | $97.9 \pm 1.5$ | $0.196 \pm 0.108$ | $96.9 \pm 3.0$ | $0.214 \pm 0.141$ |
| Spread | $99.0 \pm 0.9$ | $0.162 \pm 0.144$ | $98.6 \pm 1.3$ | $0.171 \pm 0.128$ |
| Formation | $98.3 \pm 1.0$ | $0.123 \pm 0.940$ | $98.3 \pm 1.8$ | $0.126 \pm 0.100$ |
| Line | $98.6 \pm 0.5$ | $0.117 \pm 0.540$ | $98.3 \pm 0.5$ | $0.121 \pm 0.630$ |
| Corridor | $97.9 \pm 1.8$ | $0.247 \pm 0.390$ | $98.6 \pm 1.9$ | $0.255 \pm 0.470$ |
| ConnectSpread | $97.9 \pm 1.7$ | $0.324 \pm 0.187$ | $99.0 \pm 0.8$ | $0.339 \pm 0.201$ |

Table 2: Effect of varying $\xi$ (Section 4.3) for Line (N=3) with fixed $\nu = 0.5$.

| $\xi$ | 0.5 | 0.4 | 0.2 | 0.0 |
|---|---|---|---|---|
| Safety rate | $100.0 \pm 0.0$ | $98.6 \pm 0.5$ | $96.5 \pm 0.5$ | $93.4 \pm 0.020$ |
| Cost | $0.127 \pm 0.061$ | $0.117 \pm 0.540$ | $0.108 \pm 0.044$ | $0.102 \pm 0.035$ |

**Varying $\xi$ in the outer problem** To robustify our approach against estimation errors in $V^h$, we solve for a $z_i$ that is slightly more conservative by modifying (15b) to $V_\psi^h(o_i, z_i) \leq -\xi$ (Section 4.3). We now perform experiments to study the effect of different choices of $\xi$ (Table 2). The results show that higher values of $\xi$ result in higher safety rates and slightly higher costs, while the reverse is true for smaller $\xi$. This matches our intuition that modifying (15b) can help improve constraint satisfaction when the learned $V^h$ has estimation errors. We thus recommend choosing $\xi$ close to $\nu$.

# 6 CONCLUSION

This paper introduces EFMARL for the multi-agent constrained optimal control problem. EFMARL extends the epigraph form method from single-agent RL to MARL and addresses the training insta-bility of Lagrangian methods in the zero-constraint violation setting. We decompose the epigraph form problem into a centralized inner problem solved in centralized training, and a distributed outer problem solved during online execution. Experimental results on MPE and the safe Multi-agent MuJoCo suggest that, unlike baseline methods, EFMARL uses a constant set of hyperparameters across all environments, and achieves a safety rate similar to the most conservative baseline and similar performance to the baselines that prioritize performance but violate safety constraints.

**Limitations** The theoretical analysis in Section 4.3 suggests that the connected agents must com-municate $z$ and reach a consensus. If the communication on $z$ is disabled, although our experiments show that the agents still perform similarly, the theoretical optimality guarantee may not be valid. In addition, the framework does not consider noise, disturbances in the dynamics, or communication delays between agents. Finally, as a safe RL method, although safety can be theoretically guaranteed under the optimal value function and policy, this does not hold under inexact minimization of the losses. We leave tackling these issues as future work.

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

## A PROOF OF PROPOSITION 1

*Proof.* Under the dynamics $x^{k+1} = f(x^k, \pi(x^k))$, we have

$$V(x^k, z^k; \pi) = \min_{u^k} \max \left\{ \max_{p \geq k} h(x^p), \sum_{p \geq k} l(x^p, \pi(x^p)) - z^k \right\}$$

$$= \min_{u^k} \max \left\{ \max\{h(x^k), \max_{p \geq k+1} h(x^p)\}, \sum_{p \geq k+1} l(x^p, \pi(x^p)) + l(x^k, \pi(x^k)) - z^k \right\}$$

$$= \min_{u^k} \max \left\{ \max\{h(x^k), \max_{p \geq k+1} h(x^p)\}, \sum_{p \geq k+1} l(x^p, \pi(x^p)) - \underbrace{\left[ z^k - l(x^k, \pi(x^k)) \right]}_{:= z^{k+1}} \right\}$$

$$= \min_{u^k} \max \left\{ h(x^k), \max \left\{ \max_{p \geq k+1} h(x^p), \sum_{p \geq k+1} l(x^p, \pi(x^p)) - z^{k+1} \right\} \right\}$$

$$= \min_{u^k} \max \left\{ h(x^k), V(x^{k+1}, z^{k+1}; \pi) \right\},$$

$$\tag{16}$$

where we have defined $z^{k+1} = z^k - l(x^k, \pi(x^k))$ in the third equation. $\square$

## B PROOF OF THEOREM 1

To prove Theorem 1, first, we prove several lemmas:

**Lemma 1.** *For any fixed state $x$, let $z^*$ denote the solution of (15), i.e.,*

$$\min_z \quad z, \tag{17a}$$

$$\text{s.t.} \quad V^h(x; \pi(\cdot, z)) \leq 0, \tag{17b}$$

*and let $\pi^*$ denote $\pi(\cdot, z^*)$, i.e., it is the optimal policy for $z^*$:*

$$\pi^* = \arg\min_\pi V(x, z^*; \pi). \tag{18}$$

*Then, no other safe policy $\tilde{\pi}$ exists that has a strictly lower cost than $\pi^*$ while satisfying the constraints, i.e.,*

$$V^h(x; \tilde{\pi}) \leq 0 \tag{19a}$$

$$V^l(x; \tilde{\pi}) < V^l(x; \pi^*). \tag{19b}$$

*In other words, $\pi^*$ is the optimal solution of the original constrained optimization problem*

$$\min_\pi \quad V^l(x; \pi), \tag{20a}$$

$$\text{s.t.} \quad V^h(x; \pi) \leq 0. \tag{20b}$$

Before proving this lemma, we first prove the following lemma.

**Lemma 2.** *Suppose that such a $\tilde{\pi}$ exists. Then, there exists a $z^\dagger := V^l(x; \tilde{\pi}) - V^h(x; \tilde{\pi})$ for which the optimal policy $\pi^\dagger$ for $z^\dagger$ satisfies the conditions for $\tilde{\pi}$ in (19a) and (19b), i.e.,*

$$V^h(x; \pi^\dagger) \leq V^h(x; \tilde{\pi}) \leq 0, \tag{21a}$$

$$V^l(x; \pi^\dagger) \leq V^l(x; \tilde{\pi}) < V^l(x; \pi^*). \tag{21b}$$

*Proof.* Since $\pi^\dagger$ is optimal for $z^\dagger$, we have that

$$V(x, z^\dagger; \pi^\dagger) \leq V(x, z^\dagger; \tilde{\pi}). \tag{22}$$

This implies that, by definition of $z^\dagger$,

$$\max\left\{V^h(x; \pi^\dagger),\ V^l(x; \pi^\dagger) - z^\dagger\right\} \leq \max\left\{V^h(x; \tilde{\pi}),\ V^l(x; \tilde{\pi}) - z^\dagger\right\}, \tag{23}$$

$$= V^h(x; \tilde{\pi}). \tag{24}$$

In particular,

$$V^h(x; \pi^\dagger) \leq V^h(x, \tilde{\pi}), \tag{25}$$

and

$$V^l(x; \pi^\dagger) - \left(V^l(x; \tilde{\pi}) - V^h(x; \tilde{\pi})\right) \leq V^h(x, \tilde{\pi}), \tag{26}$$

$$\implies V^l(x; \pi^\dagger) \leq V^l(x; \tilde{\pi}). \tag{27}$$

which proves (21a) and (21b). $\qquad \square$

We are now ready to prove Lemma 1.

*Proof of Lemma 1.* We prove this by contradiction.

Suppose that such a $\tilde{\pi}$ exists. By Lemma 2, there exists $z^\dagger$ and $\pi^\dagger$ that satisfies the conditions for $\tilde{\pi}$ in (19a) and (19b). Since $\pi^*$ is optimal for $z^*$, this implies that

$$\max\left\{V^h(x; \pi^*),\ V^l(x; \pi^*) - z^*\right\} \leq \max\left\{V^h(x; \pi^\dagger),\ V^l(x; \pi^\dagger) - z^*\right\}. \tag{28}$$

We now consider two cases depending on the value of the $\max$ on the right.

**Case 1** ($V^h(x; \pi^\dagger) \leq V^l(x; \pi^\dagger) - z^*$)**:** For this case, $\max\left\{V^h(x; \pi^\dagger),\ V^l(x; \pi^\dagger) - z^*\right\} = V^l(x; \pi^\dagger) - z^*$. This implies that

$$V^l(x; \pi^*) - z^* \leq V^l(x; \pi^\dagger) - z^* \qquad \Longleftrightarrow \qquad V^l(x; \pi^*) \leq V^l(x; \pi^\dagger). \tag{29}$$

However, this contradicts our assumption that $V^l(x; \pi^\dagger) \leq V^l(x; \tilde{\pi}) < V^l(x; \pi^*)$ from (19b).

**Case 2** ($V^h(x; \pi^\dagger) > V^l(x; \pi^\dagger) - z^*$)**:** For this case, $\max\left\{V^h(x; \pi^\dagger),\ V^l(x; \pi^\dagger) - z^*\right\} = V^h(x; \pi^\dagger)$. This implies that

$$V^h(x; \pi^*) \leq V^h(x; \pi^\dagger) \tag{30}$$

and

$$V^l(x; \pi^*) - z^* \leq V^h(x; \pi^\dagger) \qquad \implies \qquad V^l(x; \pi^*) - V^h(x; \pi^\dagger) \leq z^*. \tag{31}$$

However, if we examine the definition of $z^\dagger$, we have that

$$z^\dagger = V^l(x; \tilde{\pi}) - V^h(x; \tilde{\pi}) \tag{32}$$

$$\leq V^l(x; \tilde{\pi}) - V^h(x; \pi^\dagger) \qquad \text{(from (21a) and (30))} \tag{33}$$

$$< V^l(x; \pi^*) - V^h(x; \pi^\dagger) \qquad \text{(from (19b))} \tag{34}$$

$$\leq z^* \qquad \text{(from (31))}. \tag{35}$$

This contradicts our definition of $z^*$ being the optimal solution of (17), since $z^\dagger$ satisfies $V^h(x; \pi(\cdot, z^\dagger)) \leq 0$ but is also strictly smaller than $z^*$.

Since both cases lead to a contradiction, no such $\tilde{\pi}$ can exist. $\qquad \square$

We can now prove Theorem 1, which follows as a consequence of Lemma 1.

*Proof of Theorem 1.* Since $V^h(x; \pi) = \max_i V^h_i(x_i, o_i; \pi)$, Lemma 1 implies that Equation (13) is equivalent to

$$\min_z \quad z \tag{36a}$$

$$\text{s.t.} \quad V^h_i(x_i, o_i; \pi(\cdot, z)) \leq 0, \quad \forall i. \tag{36b}$$

This is equivalent to (15). □

## C    DISCUSSION ON IMPORTANCE OF PROPOSITION 1

Establishing Proposition 1 is *key* to EFMARL. Namely,

1. Satisfying dynamic programming implies that the value function is *Markovian*. In other words, for a given $z^0$, the value at the $k$th timestep is *only* a function of $z^k$ and $x^k$ instead of the $z^0$ and the *entire* trajectory up to the $k$th timestep.

2. Consequently, this implies that the optimal policy will also be Markovian and is *only* a function of $z^k$ and $x^k$.

3. Rephrased differently, since the value function is Markovian, this implies that, for a given $z^0$ and $x^0$, the value at the $k$th timestep is *equal* to the value (at the initial timestep) of a *new* problem where we start with $\tilde{z}^0 = z^k$ and $\tilde{x}^0 = x^k$.

4. Since we relate the value function of consecutive timesteps, given a value function estimator, we can now control the bias-variance tradeoff of the value function estimate by using k-step estimates instead of the Monte Carlo estimates.

5. Instead of only using the k-step estimates for a single choice of k, we can compute a weighted average of the k-step estimates as in GAE to further control the bias-variance tradeoff.

## D    EFMARL ALGORITHM

We describe the centralized training process of EFMARL in Algorithm 1 and the decentralized execution process in Algorithm 2.

---
**Algorithm 1** EFMARL Centralized Training
---
**Initialize:** Policy NN $\pi_\theta$, cost value function NN $V^l_\phi$, constraint value function NN $V^h_\psi$.
**while** Training not end **do**
    Randomly sampling initial conditions $x^0$, and the initial $z^0 \in [z_{\min}, z_{\max}]$.
    Use $\pi_\theta$ to sample trajectories $\{x^0, \ldots, x^T\}$, with $z$ dynamics (14).
    Calculate the cost value function $V^l_\phi(x, z)$ and the constraint value function $V^h_\psi(o_i, z)$.
    Calculate GAE with the total value function (12).
    Update the value functions $V^l_\phi$ and $V^h_\psi$ using TD error.
    Update the $z$-conditioned policy $\pi_\theta(\cdot, z)$ using PPO loss.
**end while**
---

## E    EXPERIMENTS

### E.1    COMPUTATION RESOURCES

The experiments are run on a 13th Gen Intel(R) Core(TM) i7-13700KF CPU with 64GB RAM and an NVIDIA GeForce RTX 3090 GPU. The training time is around 6 hours ($10^5$ steps) for EFMARL and InforMARL-L, and around 5 hours for InforMARL.

---

**Algorithm 2** EFMARL Decentralized Execution

---

**Input:** Learned policy NN $\pi_\theta$, constraint value function NN $V_\psi^h$.
**for** $k = 0, \ldots, T$ **do**
    Get $z_i$ for each agent by solving the decentralized EF-MACOCP outer problem (15b).
    **if** $z$ communication enabled **then**
        The connected agents $j$ communicate $z_j$ and reach a consensus $z = \max_j z_j$.
        Set $z_i = z$ for all agents in the connected graph.
    **end if**
    Get decentralized policy $\pi_i(\cdot) = \pi_\theta(\cdot, z_i)$.
    Execution control $u_i^k = \pi_i(o_i^k)$.
**end for**

---

### E.2 ENVIRONMENTS

#### E.2.1 MULTI-PARTICAL ENVIRONMENTS (MPE)

We use directed graphs $\mathcal{G} = (\mathcal{V}, \mathcal{E})$ to represent MPE, where $\mathcal{V}$ is the set of nodes containing the objects in the multi-agent environment (e.g., agents $\mathcal{V}_a$, goals $\mathcal{V}_g$, landmarks $\mathcal{V}_l$, and obstacles $\mathcal{V}_o$). $\mathcal{E} \subseteq \{(i, j) \mid i \in \mathcal{V}_a, j \in \mathcal{V}\}$ is the set of edges, denoting the information flow from a sender node $j$ to a receiver agent $i$. An edge $(i, j)$ exists only if the communication between node $i$ and $j$ can happen, which means the distance between node $i$ and $j$ should be within the communication radius $R$ in partially observable environments. We define the neighborhood of agent $i$ as $\mathcal{N}_i := \{j \mid (i, j) \in \mathcal{E}\}$. The node feature $v_i$ includes the states of the node $x_i$ and a one-hot encoding of the type of the node $i$ (e.g., agent, goal, landmark, or obstacle), e.g., $[0, 0, 1]^\top$ for agent nodes, $[0, 1, 0]^\top$ for goal nodes, and $[1, 0, 0]^\top$ for obstacle nodes. The edge feature $e_{ij}$ includes the information passed between the sender node $j$ and the receiver node $i$ (e.g., relative positions and velocities).

We consider 6 MPE: Target, Spread, Formation, Line, Corridor, and ConnectSpread. In each environment, the agents need to work collaboratively to finish some tasks:

**Target** (Nayak et al., 2023): Each agent tries to reach its preassigned goal.

**Spread** (Dames et al., 2017): The agents are given a set of (not preassigned) goals to cover.

**Formation** (Agarwal et al., 2020): Given a landmark, the agents should spread evenly on a circle with the landmark as the center and a given radius.

**Line** (Agarwal et al., 2020): Given two landmarks, the agents should spread evenly on the line between the landmarks.

**Corridor**: A set of agents and goals are separated by a narrow corridor, whose width is smaller than $4r$ where $r$ is the radius of agents. The agents should go through the corridor and cover the goals.

**ConnectSpread**: A set of agents and goals are separated by a large obstacle with a diameter larger than the communication radius $R$. The agents should cover the goals without colliding with obstacles or each other while also maintaining the connectivity of all agents.

We consider $n = 3$ agents for all environments and $n = 5$ and 7 agents in the Formation and Line environments. To make the environments more difficult than the original ones (Nayak et al., 2023), we add 3 static obstacles in the first 4 environments.

In our modified MPE, the state of the agent $i$ is given by $x_i = [p_i^x, p_i^y, v_i^x, v_i^y]^\top$, where $[p_i^x, p_i^y]^\top :=$ $p \in \mathbb{R}^2$ is the position of agent $i$, and $[v_i^x, v_i^y]$ is the velocity. The control inputs are given by $u_i = [a_i^x, a_i^y]^\top$, i.e., the acceleration along each axis. The joint state is defined by concatenation: $x = [x_1; \ldots; x_N]$. The agents are modeled as double integrators with dynamics

$$\dot{x}_i = \begin{bmatrix} v_i^x & v_i^y & a_i^x & a_i^y \end{bmatrix}^\top. \tag{37}$$

The agents' control inputs are limited by $[-1, 1]$, and the velocities are limited by $[-1, 1]$. The agents have a radius $r_a = 0.05$, and the communication radius is assumed to be $R = 0.5$. The area side length $L$ is $1.0$ for the Corridor and the ConnectSpread environments and $1.5$ for other environments. The radius of the obstacles $r_o$ is $0.4$ in the Corridor environment, $0.25$ in the ConnectSpread

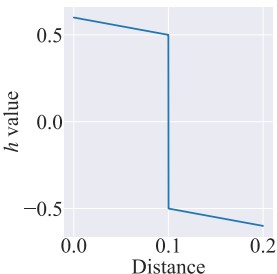

Figure 7: $h$ value with respect to distance.

environment, and $0.05$ for other environments. All environments use a simulation time step of $0.03$s and a total horizon of $T = 128$.

The observation of the agents $o_i$ includes the node features of itself, its neighbors $j \in \mathcal{N}_i$, and the edge features of the edge connecting agent $i$ and its neighbors. The node features include neighbors' states $x_j$ and its type ($[0, 0, 1]$ for agents, $[0, 1, 0]$ for goals and landmarks, and $[1, 0, 0]^\top$ for the obstacles). The edge features are the relative states $e_{ij} = x_i - x_j$.

The constraint function $h$ contains two parts for all environments except for ConnnectSpread, including agent-agent and agent-obstacles collisions. In the ConnectSpread environment, another constraint regarding the connectivity of the agent graph is considered. For the agent-agent collision, we use the $h$ function defined as

$$h_a(o_i) = 2r_a - \min_{j \in \mathcal{N}_i} \|p_i - p_j\| + \nu \mathrm{sign} \left( 2r_a - \min_{j \in \mathcal{N}_i} \|p_i - p_j\| \right), \tag{38}$$

where $\mathrm{sign}$ is the sign function, and $\nu = 0.5$ in all our experiments. This represents a linear function w.r.t. the inter-agent distance with a discontinuity at the safe-unsafe boundary (Figure 7). For the agent-obstacle collision, we use

$$h_o(o_i) = r_a + r_o - \min_{j \in \mathcal{N}_i^o} \|p_i - p_j\| + \nu \mathrm{sign} \left( r_a + r_o - \min_{j \in \mathcal{N}_i^o} \|p_i - p_j\| \right), \tag{39}$$

where $\mathcal{N}_i^o$ is the observed obstacle set of agent $i$. Then, the total $h$ function is defined as $h(o_i) = \max\{h_a(o_i), h_o(o_i)\}$ for environments except for ConnectSpread. For ConnectSpread, we also consider the connectivity constraint

$$h_c(o_i) = \max_i \min_{j \in \mathcal{N}_i^o} \|p_i - p_j\| - R' + \nu \mathrm{sign} \left( \max_i \min_{j \in \mathcal{N}_i^o} \|p_i - p_j\| - R' \right), \tag{40}$$

where $R' = 0.45$ is the required maximum distance for connected agents such that if the distance between two agents is larger than $R'$, they are considered disconnected. Note that this cost is only valid with agent number $N \leq 3$. For a larger number of agents, the second largest eigenvalue of the graph Laplacian matrix can be used. Still, since we only use this environment with 3 agents, we use this cost to decrease the complexity. Then, the total $h$ function of the ConnectSpread environment is defined as $h(o_i) = \max\{h_a(o_i), h_o(o_i), h_c(o_i)\}$.

Two types of cost functions are used in the environments. The first type is the *Target* cost used in the Target environment, which is defined as

$$l(x, u) = \frac{1}{N} \sum_{i=1}^{N} \left( 0.01 \|p_i - p_i^{\mathrm{goal}}\| + 0.001 \mathrm{sign} \left( \mathrm{ReLU}(\|p_i - p_i^{\mathrm{goal}}\| - 0.01) \right) + 0.0001 \|u_i\|^2 \right). \tag{41}$$

The first term penalizes the agents if they cannot reach the goal, the second term penalizes the agents if they cannot reach the goal exactly, and the third term encourages small controls. The second type

is the *Spread* cost used in all other environments, defined as

$$l(x,u) = \frac{1}{N}\sum_{j=1}^{N}\min_{i\in\mathcal{V}_a}\left(0.01\|p_i - p_j^{\text{goal}}\| + 0.001\text{sign}\left(\text{ReLU}(\|p_i - p_j^{\text{goal}}\| - 0.01)\right)\right.$$
$$\left. +0.0001\|u_j\|^2\right). \tag{42}$$

Instead of matching the agents to their preassigned goals, each goal finds its nearest agent and penalizes the whole team with the distance between them. In this way, the optimal policy of the agents is to cover all goals collaboratively.

### E.2.2 SAFE MULTI-AGENT MUJOCO ENVIRONMENTS

We also test on the Safe HalfCheetah(2x3) and Safe Coupled HalfCheetah(4x3) tasks from the Safe Multi-Agent Mujoco benchmark suite Gu et al. (2023). Each agent controls a subset of joints and must cooperate to maximize the reward while avoiding violating safety constraints. The task is parametrized by the two numbers in the parenthesis, where the first number denotes the number of agents, while the second number denotes the number of joints controlled by each agent. The goal for the Safe HalfCheetah and Safe Coupled HalfCheetah tasks is to maximize the forward velocity but avoid colliding with a wall in front that moves forward at a predefined velocity.

**Note:** Although this is not a homogeneous MAS, since each agent has the same control space (albeit with different dynamics), we can convert this into a homogeneous MAS by *augmenting* the state space with a one-hot vector to identify each agent, then augmenting the dynamics to use the appropriate per-agent dynamics function. **This is the approach taken in the official implementation of Safe Multi-Agent Mujoco** [6] **from Gu et al. (2023).**

For more details, see Gu et al. (2023).

### E.3 IMPLEMENTATION DETAILS AND HYPERPARAMETERS

We parameterize the $z$-conditioned policy $\pi_\theta(o_i, z)$, cost-value function $V_\phi^l(x, z)$, and the constraint-value function $V_\psi^h(o_i, z)$ using graph transformers (Shi et al., 2020) with parameters $\theta$, $\phi$, and $\psi$, respectively. Note that the policy and the constraint-value function are decentralized and take only the local observation $o_i$ as input, while the cost-value function is centralized. In each layer of the graph transformer, the node features are updated with $v_i' = W_1 v_i + \sum_{j\in\mathcal{N}_i}\alpha_{ij}(W_2 v_j + W_3 e_{ij})$, where $W_i$ are learnable weight matrices, and the $\alpha_{ij}$ is the attention weight between agent $i$ and agent $j$ computed as $\alpha_{ij} = \text{softmax}(\frac{1}{\sqrt{c}}(W_4 x_i)^\top(W_5 x_j))$, where $j$ is the first dimension of $W_i$. In this way, the observation $o_i$ is encoded. If the environment allows $M$-hop information passing, we can apply the node feature update $M$ times so that agent $i$ can receive information from its $M$-hop neighbors. After the information passing, the updated node features $v_i'$ are concatenated with the encoded $z$ vector $W_7 z$ then passed to another NN or a recurrent neural network (RNN) (Hausknecht & Stone, 2015) to obtain the outputs. $\pi_\theta$ and $V_\psi^h$ have the same structure as introduced above with different output dimensions because they are decentralized. For the centralized $V^l(x, z)$, the averaged node features after information passing are concatenated with the encoded $z$ and passed to the final layer (NN or RNN) to obtain the global cost value for the whole MAS.

When updating the neural networks, we follow the PPO (Schulman et al., 2017) structure. First, we calculate the target cost-value function $V_{\text{target}}^l$ and the target constraint-value function $V_{\text{target}}^h$ using GAE estimation (Schulman et al., 2015b), and then backpropagate the following mean-square error

---

[6]https://github.com/chauncygu/Safe-Multi-Agent-Mujoco/blob/
2e6e82c92bafd3183bf9a939fb9de35412c41d9a/safety_multi_agent_mujoco/
safety_ma_mujoco/safety_multiagent_mujoco/mujoco_multi.py#L205-L218

to update the value function parameters $\phi$ and $\psi$:

$$\mathcal{L}_{V^l}(\phi) = \frac{1}{M} \sum_{k=1}^{M} \|V_\phi^l(x^k, z^k) - V_{\text{target}}^l(x^k, z^k)\|^2, \tag{43}$$

$$\mathcal{L}_{V^h}(\psi) = \frac{1}{MN} \sum_{k=1}^{M} \sum_{i=1}^{N} \|V_\psi^h(o_i^k, z^k) - V_{\text{target}}^h(o_i^k, z^k)\|^2, \tag{44}$$

where $M$ is the number of samples. Then, we calculate the advantages $A_i$ for each agent with the total value function $V_i(x, z) = \max\{V_\phi^l(x, z) - z, V_\psi^h(o_i, z)\}$ following the same process as in PPO by replacing $V^l$ with $V$, and backpropagate the following PPO policy loss to update the policy parameters $\theta$:

$$\mathcal{L}_\pi(\theta) = \frac{1}{MN} \sum_{k=1}^{M} \sum_{i=1}^{N} \Bigg[ \min\Bigg\{ \frac{\pi_\theta(o_i^k, z^k)}{\pi_{\text{old}}(o_i^k, z^k)} A_i(x^k, z^k),$$
$$\text{clip}\left( \frac{\pi_\theta(o_i^k, z^k)}{\pi_{\text{old}}(o_i^k, z^k)}, 1 - \epsilon_{\text{clip}}, 1 + \epsilon_{\text{clip}} \right) A_i(x^k, z^k) \Bigg\} \Bigg]. \tag{45}$$

Most of the hyperparameters of EFMARL are shared with InforMARL and InforMARL-L. The values of the share hyperparameters are provided in Table 3.

Table 3: Shared hyperparameters of EFMARL, InforMARL, and InforMARL-L.

| Hyperparameter | Value | Hyperparameter | Value |
|---|---|---|---|
| policy GNN layers | 2 | RNN type | GRU |
| massage passing dimension | 32 | RNN data chunk length | 16 |
| GNN output dimension | 64 | RNN layers | 1 |
| number of attention heads | 3 | number of sampling environments | 128 |
| activation functions | ReLU | gradient clip norm | 2 |
| GNN head layers | (32, 32) | entropy coefficient | 0.01 |
| optimizer | Adam | GAE $\lambda$ | 0.95 |
| discount $\gamma$ | 0.99 | clip $\epsilon$ | 0.25 |
| policy learning rate | 3e-4 | PPO epoch | 1 |
| $V^l$ learning rate | 1e-3 | batch size | 16384 |
| network initialization | Orthogonal | layer normalization | True |

Apart from the shared hyperparameters, EFMARL has additional hyperparameters, as shown in Table 4. In addition, $z_{\min}$ and $z_{\max}$ are the lower and upper bound of $z$ while sampling $z$ in training. Since $z_{\min}$ represents an estimate of the minimum cost incurred by the MAS, we set it to a small negative number $-0.5$. We set $z_{\max}$ differently depending on the complexity of the environment. For MPE, with maximum simulation timestep $T$, we estimate it in the MPE environments using the following equation:

$$z_{\max} = \tilde{l}_{\max} * T, \tag{46}$$

$$\tilde{l}_{\max} = \text{initdist}_{\max} w_{\text{distance}} + w_{\text{reach}} + u_{\max} w_{\text{control}}, \tag{47}$$

where $\tilde{l}_{\max}$ is a conservative estimate of the maximum cost $l$. This is conservative in the sense that this reflects the case where 1) the agents and goals are initialized with the maximum possible distance ($\text{initdist}_{\max}$); 2) the agents do not reach their goal throughout their trajectory; 3) the agents incur the maximum control cost for all timesteps. $w_{\text{distance}}$, $w_{\text{reach}}$, and $w_{\text{control}}$ denote the corresponding weights of the different cost terms in the cost function $l$ in (41) and (42). For the multi-agent MuJoCo environments, we first train the agents with (unconstrained) MAPPO with different random seeds, record the largest cost incurred, double it, and then use that as $z_{\max}$.

All the hyperparameters remain the same in all environments or are pointed out in the tables except for the training steps. The training step is $10^5$ in the Target and the Spread environments, $1.5 \times 10^5$ in the Line environment, and $2 \times 10^5$ in other MPE. For the safe multi-agent MuJoCo environments, we set the training step to $7 \times 10^3$.

Table 4: Hyperparameters of EFMARL

| Hyperparameter | Value |
|---|---|
| $V^h$ GNN layers | 2 for ConnectSpread, 1 for others |
| $z$ encoding dimension | 8 |
| outer problem solver | Chandrupatla's method (Chandrupatla, 1997) |
| $z_{min}$ | -0.5 |
| $z_{max}$ | $(0.01(\sqrt{2l}) - 0.001 - 0.0001)T$ |

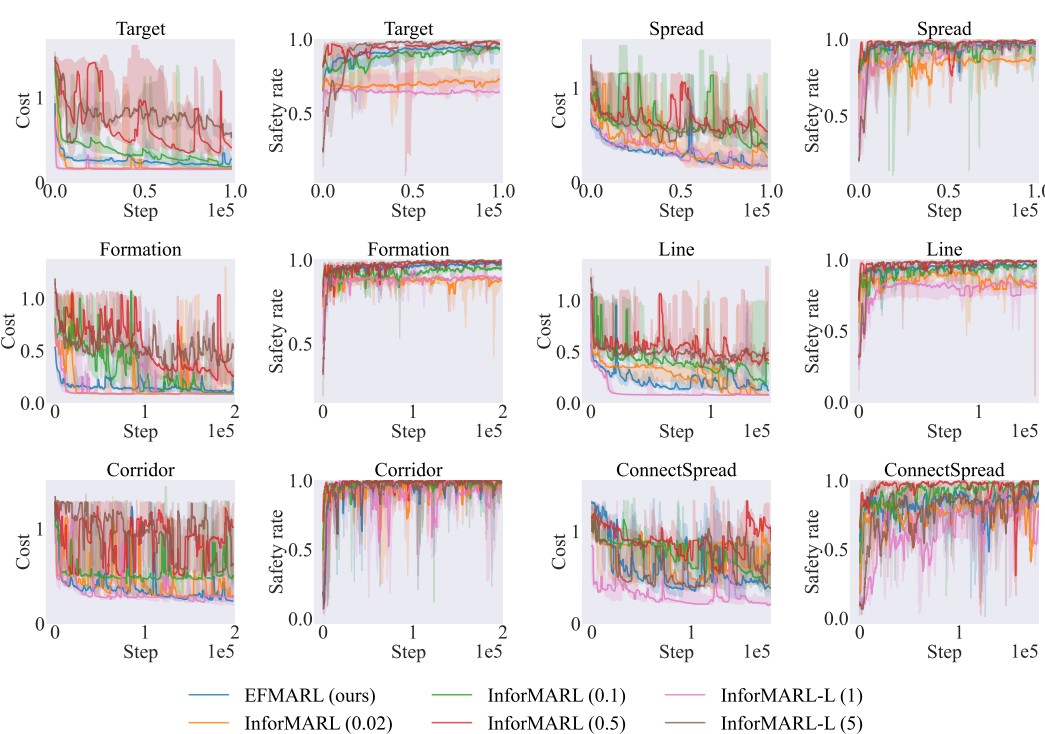

EFMARL (ours)  InforMARL (0.1)  InforMARL-L (1)
InforMARL (0.02)  InforMARL (0.5)  InforMARL-L (5)

Figure 8: Cost and safety rate of EFMARL and the baselines during training in MPE.

### E.4 IMPLEMENTATION OF THE BASLINES

The implementation of the baseline follows their original implementations:

- InforMARL: https://github.com/nsidn98/InforMARL (MIT license)

- MAPPO-L: https://github.com/chauncygu/Multi-Agent-Constrained-Policy-Optimisation (MIT License)

### E.5 TRAINING CURVES

To show the training stability of EFMARL, we have shown the cost and safety rate of EFMARL and InforMARL-L (lr) during training in the Target and Spread environments in the main pages (Figure 5). Due to page limits, we provide the plots for other environments here in Figure 8, Figure 9, and Figure 10. The figures show that EFMARL achieves stable training in all environments. Specifically, as shown in Figure 9, while InforMARL-L (lr) suffers from training instability because the constraint violation threshold is zero (as discussed in Section 3.2), EFMARL is much more stable.

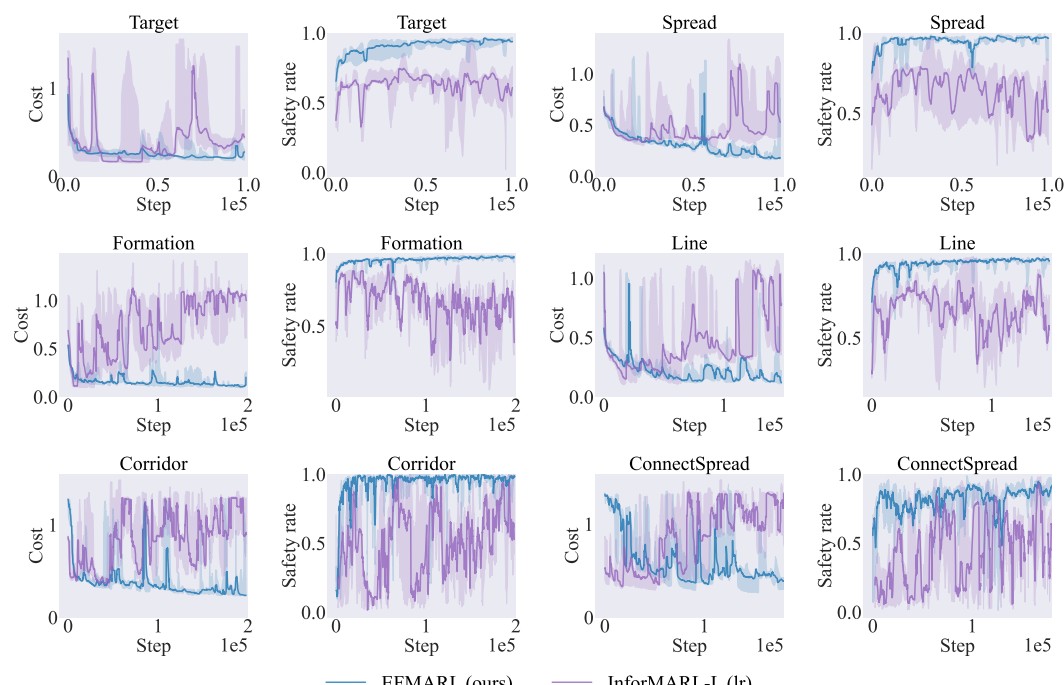

Figure 9: Cost and safety rate of EFMARL and InforMARL-L (lr) during training in MPE.

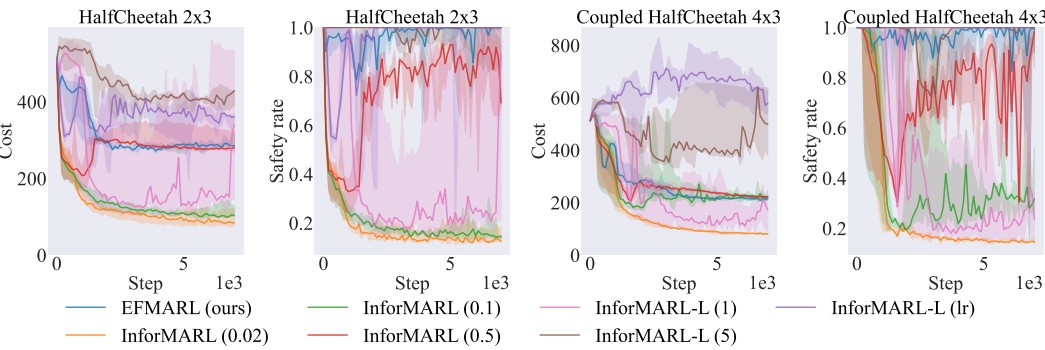

Figure 10: Cost and safety rate of EFMARL and all baselines during training in safe Multi-agent MuJoCo environments.

### E.6 MORE COMPARISON WITH THE LAGRANGIAN METHOD

In this section, we provide more comparisons between EFMARL and the Lagrangian method, where we change the constraint-value function of the Lagrangian method from the sum-over-time (SoT) form to the max-over-time (MoT) form. Using the MoT form, the constraint-value function of the Lagrangian method becomes the same as the one used in EFMARL (Equation (9)). We create 3 more baselines using this approach with different learning rates (lr) of the Lagrangian multiplier $\lambda$, where $\mathrm{lr}(\lambda) \in \{0.1, 0.2, 0.3\}$. The baselines are called InforMARL-L-MoT. We compare EFMARL with the new baselines in the Target environment, and the results are presented in Figure 11. We can observe that the Lagrangian method has very different performance with different learning rates of $\lambda$. With $\mathrm{lr}(\lambda) = 0.1$, the learned policy is unsafe, and with $\mathrm{lr}(\lambda) = 0.2$ or $0.3$, the training is unstable and the cost of the converged policy is much higher than EFMARL. In addition, we also plot the $\lambda$ values during training in Figure 11. It shows that $\lambda$ keeps increasing without converging to some value, which also suggests the instability of the Lagrangian method.

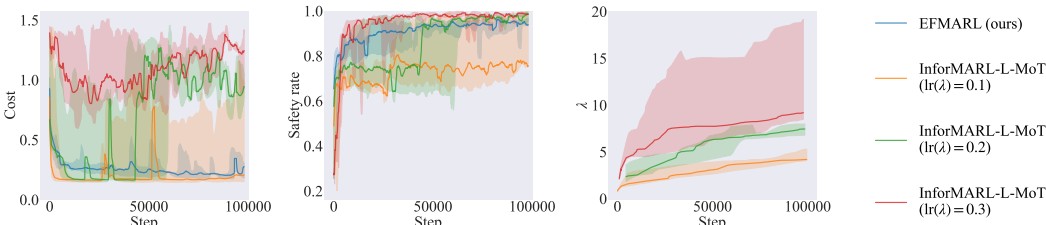

Figure 11: Cost and safety rate of EFMARL and InforMARL-L-MoT with different learning rates of $\lambda$ during training in the Target environment, and the $\lambda$ values during training.

### E.7 SENSITIVITY ANALYSIS ON THE CHOICE OF $z_{\max}$

In Appendix E.3, we have introduced how to determine the sampling interval of $z$. Here, we perform experiments in the Spread environment to study the sensitivity of EFMARL on the choice of $z_{\max}$. In this experiment, we scale the value of $z_{\max}$ used for sampling $z$, and denote by $z_{\max,\mathrm{orig}}$ the original value used in the experiments in the main pages, i.e., $z_{\max}/z_{\max,\mathrm{orig}} = 1.0$ uses the same value as in the main pages. We report the safety rates and the costs of the EFMARL policies trained with different $z_{\max}$ in Table 5. We see both safety and costs do not change much even when our estimate of the maximum cost $z_{\max}$ changes by up to $50\%$. If $z_{\max}$ is too large (e.g., $2\,z_{\max,\mathrm{orig}}$), the policy becomes too conservative because not enough samples of $z$ that are near $z^*$ are observed, reducing the sample efficiency. On the other hand, when $z_{\max}$ is too small (e.g., $0.25\,z_{\max,\mathrm{orig}}$), there may be states where the optimal $z^*$ does not fall within the sampled range. This causes the rootfinding step to be inaccurate, as $V^h$ will be queried at values of $z$ that were not seen during training, resulting in safety violations.

Table 5: Safety and Cost of EFMARL policies trained with different $z_{\max}$.

| $z_{\max}/z_{\max,\mathrm{orig}}$ | Safety rate | Cost |
|---|---|---|
| 0.25 | $93.8 \pm 2.4$ | $0.152 \pm 0.100$ |
| 0.5 | $98.0 \pm 1.4$ | $0.155 \pm 0.104$ |
| 1.0 | $99.0 \pm 0.9$ | $0.162 \pm 0.144$ |
| 1.5 | $99.0 \pm 0.0$ | $0.165 \pm 0.100$ |
| 2.0 | $99.0 \pm 0.1$ | $0.228 \pm 0.109$ |

### E.8 CODE

The code of our algorithm and the baselines are provided in the 'efmarl.zip' file in the supplementary materials.

## F CONVERGENCE

In this section, we analyze the convergence of the inner RL problem (13b) to a locally optimal policy.

Since we solve the inner RL problem (13b) in a centralized fashion, it can be seen as an instantiation of single-agent RL, but with a per-agent independent policy. Define the augmented state $\tilde{x} \in \tilde{\mathcal{X}} := \mathcal{X} \times \mathbb{R}$ as $[x,\ z]$, which follows the dynamics $\tilde{f} : \tilde{\mathcal{X}} \times \mathcal{U} \to \tilde{\mathcal{X}}$ defined as

$$\tilde{f}\big([x^k, z^k], u^k\big) = \big[f(x^k, u^k),\ z^k - l(x^k, u^k)\big]. \tag{48}$$

The inner RL problem (13b) can then be stated as

$$\min_{\pi}\quad \max_{k \geq 0} h(x^k, \pi(\tilde{x}^k)) \tag{49a}$$

$$\text{s.t.}\quad \tilde{x}^{k+1} = \tilde{f}(\tilde{x}^k, \tilde{\pi}(\tilde{x}^k)), k \geq 0. \tag{49b}$$

This is an instance of a single-agent RL *avoid* problem. Consequently, applying the results from (Yu et al., 2022b, Theorem 5.5) or (So et al., 2024, Theorem 4) gives us that the policy $\pi$ converges almost surely to a locally optimal policy.

## G ON THE EQUIVALENCE OF THE MACOCP AND ITS EPIGRAPH FORM

In Section 3.2, we state that the Epigraph form (5) of a constrained optimization problem is equivalent to the original problem (3). This has been proved in So & Fan (2023). To make this paper more self-contained, we also include the proof here.

*Proof.* For a constrained optimization problem (3), its epigraph form (Boyd & Vandenberghe, 2004, pp 134) is given by

$$\min_{\pi,z} \quad z, \tag{50a}$$
$$\text{s.t.} \quad h(\pi) \leq 0, \tag{50b}$$
$$J(\pi) \leq z, \tag{50c}$$

where $z \in \mathbb{R}$ is an auxiliary variable. Here, (50b) and (50c) can be combined, which leads to the following problem:

$$\min_{\pi,z} \quad z, \tag{51a}$$
$$\text{s.t.} \quad \max\{h(\pi), J(\pi) - z\} \leq 0. \tag{51b}$$

Using this form, So & Fan (2023, Theorem 3) shows that the minimization of $x$ can be moved into the constraint, which yields

$$\min_{z} \quad z, \tag{52a}$$
$$\text{s.t.} \quad \min_{\pi} \max\{h(\pi), J(\pi) - z\} \leq 0. \tag{52b}$$

This is the same as (5). □

