# OpenReview forum: "Distributed Epigraph Form Multi-Agent Safe Reinforcement Learning"
_ICLR.cc/2025/Conference — Submitted to ICLR 2025_

### Official Review · Reviewer_RYsH · 2024-10-27

**Soundness:** 2
**Presentation:** 1
**Contribution:** 2
**Rating:** 3
**Confidence:** 4

**Summary:**

This paper studies the safe marl problem under the zero constraint violation setting. A multi-agent constrained optimal control problem is formulated using the epigraph form technique in optimization theory. A safe marl algorithm is proposed, where graph neural networks are used to model distributed policies. Simulation results demonstrate the effectiveness of the proposed algorithm.

**Strengths:**

(1) Graph neural networks are used to model distributed policies to improve coordination performance.

(2) Extensive simulations are performed on standard marl benchmarks.

**Weaknesses:**

(1) There exist severe theoretical issues in the paper.

(2) The comparison between the proposed method and the Lagrange-based method is not sufficient.

(3) The paper is poorly written, which is very hard to follow.

More details are given in the question part.

**Questions:**

(1) Is the multi-agent system considered in the paper homogeneous or heterogeneous? It seems that the MAS is homogeneous in Section 3.1 due to the shared observation space. However, as stated in Section 4.2, the advantage decomposition method is used to train NNs, such that the MAS seems to be heterogeneous.

(2) (6) is not the correct Lagrangian for (3). Please provide the correct form of the Lagrangian for (3), and ensure the comparison between your method and the Lagrangian method is accurate. Can your method reduce computational complexity for example?

(3) Proposition 1 is wrong. Neither $h(x^k)$ nor $V(x^{k+1}, z^{k+1}; \pi)$ is a function of $u^k$. It also seems that the proposed algorithm is irrelated to this proposition.

(4) In the policy and value function update section, if each agent has its local policy and constraint-value function, then $\pi_{\theta}$ and $V_{\psi}^h$ should be respectively rewritten as $\pi_{\theta^i}$ and $V_{\psi^i}^h$. In addition, do you use MAPPO or HAPPO? The authors are suggested to incorporate the update formulas of NNs in the appendix.

(5) How to determine the sampling interval of $z$? What will happen if we choose the interval too large or too small?

(6) Why is $z$ updated during the policy and value function update process (lines 295-296)?

(7) The proof of Theorem 1 is very hard to follow. Both $V^h$ and $V^l$ are functions of $x$, but the authors seem to neglect this property. Formula (16) should be rewritten. Why is $\varepsilon$ positive (line 846)? In the proof of Lemma 1, what will happen if $\varepsilon  - \eta < \zeta < \varepsilon $? Too many redundant notations exist in the proof. If the authors are confident in the correctness of the theoretical derivation, please revise the proof carefully. Thanks.

(8) In section 4.3, the authors should explain how to obtain the optimal $z$ once the updates for both the policy and the value function are completed.

(9) It is confusing that the authors emphasize solving the outer problem during the distributed execution process. Why can’t this problem be addressed during centralized training? Sharing local $z_i$ in an all-to-all or centralized communication network doesn’t seem to increase communication cost, as $z_i$ is a scalar.

---

> ### Author Response · Authors · 2024-11-18
> **Reply (1/3)**
>
> We really appreciate the valuable questions raised, especially the questions that point out two important confusions (Q2 and Q3). We address all questions below.
>
> In brief, we:
> 1. Clarify the confusion in our propositions
> 2. Revise the proof of Theorem 1 to be more concise
> 3. Provide additional experimental results to demonstrate the influence of the sampling interval of $z$
>
> We sincerely hope that our reply addresses the reviewer's concerns.
>
> ## Detailed replies
>
> ***Q1, Q4**: Is the multi-agent system considered homogeneous or heterogeneous? Is MAPPO or HAPPO used? The authors are suggested to incorporate the update formulas of NNs in the appendix.*
>
> **A1, A4**: We consider homogeneous multi-agent systems and hence use MAPPO. We have clarified that the MAS is homogeneous in the revised version (Section 3.1). Following your suggestion, we have also included detailed update formulas for the NNs in Appendix E.3 in the revised paper.
>
> ---
>
> ***Q2 (1/2)**: (6) is not the correct Lagrangian for (3). Please provide the correct form of the Lagrangian for (3), and ensure the comparison between your method and the Lagrangian method is accurate.*
>
> **A2 (1/2)**: We apologize for the confusion. (6) is not the Lagrangian of (3). The CMDP setting has the constraint
> $$
> \sum_{k=0}^\infty c(x^k) \leq d
> $$
> The _zero_ constraint violation CMDP setting uses $c \geq 0$ and $d=0$. Since our constraint $h$ in (3) takes negative values, we can translate it to the _zero_ constraint violation setting by taking
> $$
> c(x) := \max\\{ 0, h(x) \\},
> $$
> which results in the constraint optimization problem
> $$
> \min_\pi J(\pi) \quad \mathrm{s.t.} \quad \sum_{k=0}^\infty \max\big\\{0, h(x^k) \big\\} \leq 0
> $$
> The Lagrangian of the above is then
> $$
> \max_{\lambda \geq 0} \min_\pi J(\pi) + \lambda \sum_{k=0}^\infty \max\\{0, h(x^k)\\}
> $$
> which is what we intended to mean with (6).
>
> We have corrected this and included the above explanation in the revised version in the "Comparison with the Lagrangian method" paragraph in Section 3.2.
>
> ***Q2 (2/2)**: Can your method reduce computational complexity for example?*
>
> **A2 (2/2)**: We do not reduce computational complexity.
> The training time of our method is similar to that of Lagrangian methods. While we have provided the training time of our method in Appendix D.1 in the original submission (Appendix E.1 in the revised paper), now we also add the training time of baselines in Appendix E.1 in the revised paper for comparison.
>
> ---
>
> ***Q3**: Proposition 1 is wrong. Neither $h(x^k)$ nor $V(x^{k+1},z^{k+1};\pi)$ is a function of $u^k$. It also seems that the proposed algorithm is irrelated to Proposition 1.*
>
> **A3**: The $\min_{u^k}$ in Proposition 1 is a typo. Thanks for spotting it! We have deleted the erroneous $\min_{u^k}$ from Proposition 1 in our revised paper.
>
> To answer the second part, we want to emphasize that Proposition 1 is actually **key** to our paper:
>
> 1. Satisfying dynamic programming implies that the value function is _Markovian_. In other words, for a given $z^0$, the value at the $k$-th timestep is _only_ a function of $z^k$ and $x^k$ instead of the $z^0$ and the _entire_ trajectory up to the $k$-th timestep.
> 2. Consequently, this implies that the optimal policy is also Markovian and is _only_ a function of $z^k$ and $x^k$.
> 3. Rephrased differently, since the value function is Markovian, this implies that, for a given $z^0$ and $x^0$, the value at the $k$-th timestep is _equal_ to the value (at the initial timestep) of a _new_ problem where we start with $\tilde{z}^0 = z^k$ and $\tilde{x}^0 = x^k$.
> 4. Since we relate the value function of consecutive timesteps, given a value function estimator, we can now control the bias-variance tradeoff of the value function estimate by using $k$-step estimates instead of the Monte Carlo estimates.
> 5. Instead of only using the $k$-step estimates for a single choice of $k$, we can compute a weighted average of the $k$-step estimates as in GAE to further control the bias-variance tradeoff.
>
> In the revision, we expand on our existing discussion in Appendix C to avoid future confusion. Thanks for bringing this topic!

---

> ### Author Response · Authors · 2024-11-18
> **Reply (2/3)**
>
> ***Q5 (1/2)**: How to determine the sampling interval of $z$?*
>
> **A5 (1/2)**: Thanks for the question! We sample $z$ uniformly from $[z_\mathrm{min}, z_\mathrm{max}]$. While we have briefly introduced the choice of the sampling interval of $z$ in Appendix D.3 and Table 4 in the original submission, here we provide a more detailed explanation of the choice of $z_\mathrm{min}$ and $z_\mathrm{max}$.
>
> Since $z_\mathrm{min}$ represents an estimate of the minimum cost incurred by the MAS, we set it to a small negative number $-0.5$.
> We set $z_\mathrm{max}$ differently depending on the complexity of the environment.
>
> **MPE**
>
> For $z_\mathrm{max}$, with maximum simulation timestep $T$, we estimate it in the MPE environments using the following equation:
>
> $$
> z_\mathrm{max} = \tilde{l}_\mathrm{max} * T,
> $$
>
> $$
> \tilde{l}\_\mathrm{max} = \mathrm{initdist}\_{\max} \\, w_\mathrm{distance} + w_\mathrm{reach} + u_\mathrm{max} w_\mathrm{control}
> $$
>
> where $\tilde{l}_\mathrm{max}$ is a conservative estimate of the maximum cost $l$. This is conservative in the sense that this reflects the case where
> - the agents and goals are initialized with the maximum possible distance ($\mathrm{initdist}_{\max}$)
> - the agents do not reach their goal throughout their trajectory
> - the agents incur the maximum control cost for all timesteps
>
> $w_\mathrm{distance}$, $w_\mathrm{reach}$, and $w_\mathrm{control}$ denote the corresponding weights of the different cost terms in the cost function $l$ in (62) and (63) of the original submission.
>
> **Multi-agent MuJoCo**
>
> For the multi-agent MuJoCo environments, we first train the agents with (unconstrained) MAPPO with different random seeds, record the largest cost incurred, double it, and then use that as $z_\mathrm{max}$.
>
> We have added the above discussion in Appendix E.3 in the revised paper.
>
> ---
>
> ***Q5 (2/2)**:  What will happen if we choose the interval too large or too small?*
>
> **A5 (2/2)**: Good question. We perform an additional experiment in the Spread environment where we scale the value of $z_\mathrm{max}$ used for sampling $z$, and denote by $z_\mathrm{max, orig}$ the original value used in the paper, i.e., $z_\mathrm{max} / z_\mathrm{max, orig} = 1.0$ uses the same value as in the paper.
>
> |$z_\mathrm{max} / z_\mathrm{max, orig}$|Safety rate|Cost|
> |---|---|---|
> |$0.25$|$93.8\pm2.4$|$0.152\pm0.100$|
> |$0.5$|$98.0\pm1.4$|$0.155\pm0.104$|
> |$1.0$|$99.0\pm0.9$|$0.162\pm0.144$|
> |$1.5$|$99.0\pm0.0$|$0.165\pm0.100$|
> |$2.0$|$99.0\pm0.1$|$0.228\pm0.109$|
>
> We see both safety and costs do not change much even when our estimate of the maximum cost $z_\mathrm{max}$ changes by up to $50\\%$.
> However, if $z_\mathrm{max}$ is too large (e.g., $2 z_\mathrm{max, orig}$), the policy becomes too conservative because not enough samples of $z$ that are near $z^*$ are observed, reducing the sample efficiency.
> On the other hand, when $z_\mathrm{max}$ is too small (e.g., $0.25 z_\mathrm{max, orig}$), there may be states where the optimal $z^*$ does not fall within the sampled range. This causes the rootfinding step to be inaccurate, as $V^h$ will be queried at values of $z$ that were not seen during training, resulting in safety violations.
>
> We have added the new experimental results and the discussion on the choice of sampling interval for $z$ in Appendix E.7.

---

> > ### Author Response · Authors · 2024-11-18
> > **Reply (3/3)**
> >
> > ***Q6**: Why is $z$ updated during the policy and value function update process (lines 295-296)?*
> >
> > **A6**: The "updating" of $z$ occurs during the trajectory rollout process by treating it as an augmented state and using (13) in the original submission for its dynamics.
> > This comes from Proposition 1, which relates the problem (and hence the value function and policy) at consecutive timesteps by considering a different value of $z$, described in (13).
> > We have updated our revision to make this part more clear ("Policy and value function updates" paragraph in Section 4.2)
> >
> > ---
> >
> > ***Q7**: The proof of Theorem 1 is very hard to follow. If the authors are confident in the correctness of the theoretical derivation, please revise the proof carefully. Thanks.*
> >
> > **A7**: Thank you for the suggestion! We agree that the proof of Theorem 1 is hard to follow and **have come up with a new concise proof** in the revised version (Appendix B) that is easier to verify the correctness of.
> >
> > ---
> >
> > ***Q8**: In section 4.3, the authors should explain how to obtain the optimal $z$ once the updates for both the policy and the value function are completed.*
> >
> > **A8**: Thanks for the suggestion! While this is stated very briefly in the Appendix already (Appendix D.3 of the original version), we have added an explanation in the main text of the revised version (Section 4.3).
> >
> > In short, we follow [1] and use an improved version of the bisection algorithm [2] to find the value of $z$ where $V^h = 0$.
> >
> > ---
> >
> > ***Q9 (1/2)**: It is confusing that the authors emphasize solving the outer problem during the distributed execution process. Why can’t this problem be addressed during centralized training?*
> >
> > **A9 (1/2)**: The outer problem cannot be addressed during centralized training because it needs the **converged** constraint value function $V^h$ and also the **current** states of agents. The optimal $z$ value changes if the states of agents change. Therefore, only when the agents know their states during execution can we determine the optimal $z$.
> >
> > ---
> >
> > ***Q9 (2/2)**: $z$ is a scalar, so communicating its value in an all-to-all or centralized network is cheap.*
> >
> > **A9 (2/2)**: This is correct. However, in our problem setting where each agent has a limited communication radius, the multi-agent system may not be connected, i.e., there may be agents that are outside the communication radius of all other agents, making such an all-to-all or centralized network infeasible.
> >
> > ## References
> >
> > [1] Oswin So and Chuchu Fan. Solving stabilize-avoid optimal control via epigraph form and deep
> > reinforcement learning. In Proceedings of Robotics: Science and Systems, 2023.
> >
> > [2] Chandrupatla, Tirupathi R. "A new hybrid quadratic/bisection algorithm for finding the zero of a nonlinear function without using derivatives." Advances in Engineering Software 28.3, 1997.

---

> > > ### Comment · Reviewer_RYsH · 2024-11-24
> > >
> > > Thanks for the responses. Some additional comments are as follows:
> > >
> > > (1) Based on the responses to Q2 and Q5, we find that the proposed epigraph form-based method can be unpractical in real-world scenarios due to the challenge of determining the sampling interval of the auxiliary variable $z$ (sometimes we even need to train the agents with some unsafe MARL algorithms to determine this interval). Additionally, the manuscript does not clarify whether
> > > $z$ has any physical meaning.
> > >
> > > (2) In Remark 1 and the proof of Theorem 1, the authors repeatedly attempt to justify that the maximum function (see the constraint in (13b)) can be replaced by a single function. We believe this replacement is not theoretically rigorous. Nevertheless, the main theoretical result of this work are built on this observation. In the proof of the theorem, a lot of new optimization problems which are different from the original problem are proposed and discussed. As a result, the contribution of this work is still questionable.
> > >
> > > (3) Another serious problem is that the authors replace the initial state distribution in the constraint of the original problem (see 5(b)) with an arbitrary state distribution in (15b). Based on the arbitrary state distribution, the authors state that they need to solve the outer problem in a distributed manner, which becomes one key contribution of this work. However, if the agents are initialized based on the initial distribution, there is no need to solve the outer problem in the distributed manner.
> > >
> > > (4) By checking the update formulas in Appendix E.3, we find that the advantage decomposition method is not employed in this work. It seems that only the single-agent PPO algorithm is used to update policy parameters.
> > >
> > > (5) The authors state that homogeneous agents are considered in this work. However, the agents in the safe Multi-agent MuJoCo environment see to be heterogeneous.

---

> ### Author Response · Authors · 2024-11-24
> **Author Reply (1/2)**
>
> Thanks for the additional comments.
>
> ***Q1 (1/2)**: Based on the responses to Q2 and Q5, we find that the proposed epigraph form-based method can be unpractical in real-world scenarios due to the challenge of determining the sampling interval of the auxiliary variable $z$ (sometimes we even need to train the agents with some unsafe MARL algorithms to determine this interval).*
>
> **R1 (1/2)**: The sampling interval of $z$ is a hyperparameter. However, since $z$ is the desired cost upper bound, the heuristics we propose provide very good estimates of this hyperparameter which **require at most one run of a MARL algorithm**.
>
> In contrast, **other safe MARL algorithms require much more than one run** to find good performing hyperparameters (e.g., static Lagrange multiplier, Lagrange multiplier learning rate).
>
> From this perspective, we argue that our proposed method is more practical than existing safe MARL methods.
>
> ---
>
> ***Q1 (2/2)**: The manuscript does not clarify whether $z$ has any physical meaning.*
>
> **R1 (2/2)**: We have mentioned multiple times in the paper that $z$ is a **desired cost upper bound** (e.g., Line 191, 220, 221). Hence, $z$ has the same units as the cost. For example, if the objective is to minimize the sum of the distance to the goal (meters), then $z$ has units of meters.
>
> Line 183 and Line 190 provide an explanation of why $z$ can be interpreted as a cost upper bound.
>
> In contrast, the Lagrange multiplier has units of (objective units) / (constraint units).
> **We believe it is easier to interpret units of the objective than it is to interpret the ratio of objective units to constraint units.**
>
> ---
>
> ***Q2 (1/2)**: In Remark 1 and the proof of Theorem 1, the authors repeatedly attempt to justify that the maximum function (see the constraint in (13b)) can be replaced by a single function. We believe this replacement is not theoretically rigorous.*
>
> **R2 (1/2)**: Thanks for the feedback! We agree our statements in Remark 1 could be more rigorous and have fixed this in the revised version.
>
> Specifically, instead of using notations that could be misinterpreted as equality of _functions_ (e.g., $V = V^l$ or $V = V^h$), we clarify that we mean that the _function evaluations_ at certain arguments are equal, e.g., for a specific $x$, $z$, and $\pi$ that
> $$
>     V(x, z; \pi) = V^l(x; \pi) \qquad \text{ or } \qquad V(x, z; \pi) = V^h(x; \pi).
> $$
>
> However, we **disagree** that the proof of Theorem 1 is not theoretically rigorous.
> **If the reviewer is confident that the proof of Theorem 1 is not rigorous, please point us to a specific line in the proof that we may have overlooked.**
>
>
> ---
>
> ***Q2 (2/2)**: In the proof of Theorem 1, a lot of new optimization problems which are different from the original problem are proposed and discussed. As a result, the contribution of this work is still questionable.*
>
> **R2 (2/2)**: In the proof of Theorem 1, we first prove Lemma 1, which is the single-agent version of Theorem 1. We then show that Lemma 1 (single-agent version) implies Theorem 1 (multi-agent version). Only **one** optimization problem (17) is introduced, in Lemma 1. We do not believe that introducing a new optimization problem in our proof makes the contribution of the paper questionable.
>
>
> ---
>
> ***Q3**: Another serious problem is that the authors replace the initial state distribution in the constraint of the original problem (see 5(b)) with an arbitrary state distribution in (15b). Based on the arbitrary state distribution, the authors state that they need to solve the outer problem in a distributed manner, which becomes one key contribution of this work. However, if the agents are initialized based on the initial distribution, there is no need to solve the outer problem in the distributed manner.*
>
> **R3**: We kindly remind the reviewer that we consider **deterministic** dynamics (1).
> Moreover, the problem we wish to solve (2) is **conditioned on $x^0$**, and **we wish to solve (2) over all $x^0$**.
> Consequently, all the optimization problems that we solve are **also** conditioned on $x^0$.
>
> In particular, (5) is conditioned on $x^0$. This means **the solution $z^\*$ to (5) will be different for different $x^0$**.
> Since the $x^0$ used during test time is not known in advance, we can not solve (5) in advance.
>
> Finally, there is **no** initial state distribution in our problem,
> and we have not mentioned the word "distribution" at all in the entire paper.

---

> > ### Author Response · Authors · 2024-11-24
> > **Author Reply (2/2)**
> >
> > ***Q4**: By checking the update formulas in Appendix E.3, we find that the advantage decomposition method is not employed in this work. It seems that only the single-agent PPO algorithm is used to update policy parameters.*
> >
> > **R4**: First: the "advantage decomposition" used in [1] (which we refer to on line 304) is **completely different** from the "value decomposition" used in off-policy methods such as QMIX.
> > **This might have been a misunderstanding.**
> >
> > Second: multi-agent PPO [2] **does not use value decomposition**.
> > The _only_ difference between **single-agent** PPO and **multi-agent** PPO is that
> > 1. **Each** agent has their own policy loss function
> > 2. The importance sampling ratio $\pi_{i, \theta}(u_i | o_i) / \pi_{i, \theta_{\text{old}}}(u_i | o_i)$ computes the density of agent $i$'s actions and **not** the density of the joint action
> >
> > These differences are justified by the "advantage decomposition" from [1].
> > Specifically, [1] says that summing up per-agent advantages computed by assuming that the other agents' policies are fixed is _equal_ to the total advantage.
> >
> > Since our update formulas in Appendix E.3 follow 1 and 2 above, we are using multi-agent PPO instead of single-agent PPO.
> >
> > To avoid possible confusion in the future, we have removed the mention to advantage decomposition in our revised manuscript because this is not the main point of our work.
> >
> > ---
> >
> > ***Q5**: The authors state that homogeneous agents are considered in this work. However, the agents in the safe Multi-agent MuJoCo environment see to be heterogeneous.*
> >
> > **R5**: Good observation! We used the same technique as the official implementation of the safe Multi-agent MuJoCo environment to **convert it to a homogeneous system**. We have added the clarification below in Appendix E.2.2:
> >
> > > **Note**: Although this is not a homogeneous MAS, since each agent has the same control space (albeit with different dynamics), we can convert this into a homogeneous MAS by **augmenting** the state space with a one-hot vector to identify each agent, then augmenting the dynamics to use the appropriate per-agent dynamics function. **This is the approach taken in the official implementation of Safe Multi-Agent Mujoco from [1].**
> >
> > For the detailed implementation of the one-hot identification vector, please refer to the original code: [https://github.com/chauncygu/Safe-Multi-Agent-Mujoco/blob/2e6e82c92bafd3183bf9a939fb9de35412c41d9a/safety_multi_agent_mujoco/safety_ma_mujoco/safety_multiagent_mujoco/mujoco_multi.py\#L205-L218](https://github.com/chauncygu/Safe-Multi-Agent-Mujoco/blob/2e6e82c92bafd3183bf9a939fb9de35412c41d9a/safety_multi_agent_mujoco/safety_ma_mujoco/safety_multiagent_mujoco/mujoco_multi.py\#L205-L218)
> >
> > ## References
> >
> > [1] Gu, Shangding, et al. "Safe multi-agent reinforcement learning for multi-robot control." Artificial Intelligence 319 (2023): 103905.
> >
> > [2] Chao Yu, Akash Velu, Eugene Vinitsky, Jiaxuan Gao, Yu Wang, Alexandre Bayen, and Yi Wu. The
> > surprising effectiveness of ppo in cooperative multi-agent games. Advances in Neural Information
> > Processing Systems, 35:24611–24624, 2022a.

---

> > > ### Author Response · Authors · 2024-12-01
> > > **We are sincerely looking forward to the reply from Reviewer RYsH**
> > >
> > > Dear Reviewer RYsH,
> > >
> > > Thank you again for carefully reading our rebuttal and asking additional questions.
> > > We have addressed them in our previous replies accordingly and have made the corresponding changes in the paper. Your questions have greatly improved the rigor and clarity of our work!
> > >
> > > In summary, we have:
> > >
> > > 1. Explained that, while the sampling interval of $z$ is a hyperparameter, we provide an efficient way of estimating a good-performing interval that requires much fewer runs than a standard hyperparameter search
> > > 2. Clarified that the physical meaning of $z$ is the desired cost upper bound
> > > 3. Replaced the equality of functions in Remark 1 with the equality of function evaluations to avoid possible misunderstandings
> > > 4. Clarified why $z$ needs to be solved online and that there is no notion of "initial state distribution" in our problem
> > > 5. Clarified that we use multi-agent PPO instead of single-agent PPO
> > > 6. Clarified that the safe multi-agent MuJoCo environment is treated as a homogeneous multi-agent system, as in the original author's codebase.
> > >
> > > As the deadline for the discussion period is approaching, we sincerely would like to ask if our reply has addressed your concerns.
> > > **If so, would you please kindly raise your score?**
> > > We are also happy to answer any further questions.

---

> > > > ### Author Response · Authors · 2024-12-02
> > > > **Kind reminder of the discussion period deadline**
> > > >
> > > > Dear Reviewer RYsH,
> > > >
> > > > Thank you again for carefully reading our rebuttal and asking additional questions.
> > > >
> > > > We kindly remind you that **the discussion period ends today** (Dec 2nd at midnight AoE). Could you **please review our reply to your additional questions and let us know if they have addressed your concerns?**
> > > > If so, would you please kindly raise your score?
> > > >
> > > > Your questions have greatly improved the rigor and clarity of our work. Your opinion on our work means a lot to us, and Reviewer $\color{#FFB000}{\textsf{tD17}}$ is also looking forward to your reply so they can make a decision.
> > > >
> > > > If you have any further questions, please do not hesitate to ask us! We are more than happy to answer them.
> > > >
> > > > Sincerely,
> > > >
> > > > Authors

---

### Official Review · Reviewer_tD17 · 2024-11-02

**Soundness:** 2
**Presentation:** 3
**Contribution:** 2
**Rating:** 5
**Confidence:** 3

**Summary:**

The paper introduces a new multi-agent constrained optimal control formulation, addressing the zero constraint violation setting. Specifically, the authors highlight and address the main limitations of Constrained MDP-based approaches in ensuring safety, particularly under strict constraints where any violation of safety requirements is unacceptable. Hence, they propose to extend the epigraph form method proposed for a single-agent scenario in (So\&Fan 2023) to multi-agent reinforcement learning (MARL), presenting Epigraph Form MARL (EFMARL).

**Strengths:**

The paper introduces a novel epigraph formulation to solve the multi-agent constrained optimal control problem. While based on established concepts, the authors' extension to the multi-agent reinforcement learning (MARL) context is not trivial. I particularly appreciate the authors' comparison to existing theoretical approaches, highlighting where current methods fail to handle constraint satisfaction and safety issues in MARL settings. The proposed formulation for the multi-agent optimal control seems an interesting and valid alternative to the well-studied Constrained MDP.

**Weaknesses:**

Even though the paper tries to address a very important problem with an elegant novel formulation, several concerns arise when reading the paper.

**On the comparison with existing literature**:

In the introduction section, the authors state that there is a lack of theoretical and practical solutions for safe RL, especially when policies are executed in a distributed manner. This statement is partially incorrect as several works, for instance, [A, B, C, D], proposed an in-depth study and benchmarks on the mentioned problem. A brief discussion and an empirical comparison against one or two of these methods are missing.

**On the global guarantees of epigraph-based solution**:

The authors correctly notice that existing Constrained MARL approaches provide safety guarantees either in the form of asymptotic convergence guarantees to the optimal safe solution or other approximate forms. Nonetheless, what type of guarantees they can provide with their novel formulation remains unclear. If I understand correctly, by using distributed constraint-value functions during execution, the agents theoretically ensure safety without needing a centralized update. In fact, as reported by the authors in lines 310-318, with this formulation, even without communication between agents, safety is ensured as long as each agent solves the local problem for each $z_i$. However, this decentralized approach primarily guarantees "constraint satisfaction" rather than task completion, meaning that while the agents may stay safe, they might not achieve the global objective if staying in place satisfies the constraints without addressing the task goals. A clear discussion of this aspect is needed to properly assess the value of the work.

**Weak empirical evaluation**:

A significant limitation of this work is the lack of a thorough empirical evaluation that addresses both task completion and scalability challenges. The primary metrics presented—cost and safety rate—demonstrate the method’s effectiveness in meeting constraints, but they do not reveal whether the agents actually complete the intended tasks. Essential performance indicators, such as mean reward (or mean success rate), are needed to clarify the method’s ability to balance constraint satisfaction with successful task completion. Without these measures, the empirical results do not fully validate the theoretical concerns raised for the CMDP formulation. For instance, a trained CMDP-based policy that opts to take no action (e.g., zero velocity) could theoretically match an epigraph-based method in having zero constraint violations, but only the latter may achieve task completion while also respecting safety constraints. Thus, without a metric for task completion, the reported experiments do not provide a comprehensive assessment of the novel formulation’s effectiveness compared to the existing formulation.

Additionally, a detailed comparison with recent Safe MARL approaches (e.g., [A, B, C]), even if they do not strictly enforce zero constraint violations, is missing. Finally, a discussion on the scalability limitations associated with using Graph Neural Networks (GNNs) as the backbone for $z$-conditioned policies is also not discussed. Although GNNs are effective in handling complex inter-agent dependencies, they may introduce computational and structural limitations that impact scalability in larger, more complex multi-agent scenarios. An in-depth investigation of these scalability issues would better validate the robustness and applicability of the proposed formulation.


**Minor comments**:

The formulation lacks a little bit of clarity. For instance, in Eq.(2), the cumulative cost over an infinite horizon is defined with $l(x^k, \pi(x^k))$, while in Eq.(3), the authors introduce the epigraph form using a "cost" function $J(x)$. While this is done only for practical purposes, however consistency in the background formulation would be beneficial to the clarity of the paper. In Eq. (7), there is $k \geq \tau$, but $\tau$ is never defined/discussed.
In line 235, the authors refer to a "total value function," which should properly be a "total \textit{cost} value function". Moreover, using the term "value function"  in an RL-based context could be misleading, as we typically denote by "value function" $V_\pi(x)$ the value of a state, considering the immediate reward plus the future discounted reward following the current policy $\pi$.

**References**:

[A] Sun, Mingfei, et al. "Trust region bounds for decentralized ppo under non-stationarity." AAMAS 2023.

[B] Wang, Z., Du, Y., Sootla, A., Ammar, H. B., \& Wang, J. (2022). CAMA: A New Framework for Safe Multi-Agent Reinforcement Learning Using Constraint Augmentation.

[C] Chen, Ziyi, Yi Zhou, and Heng Huang. "On the Duality Gap of Constrained Cooperative Multi-Agent Reinforcement Learning" (2024).

[D] Wang, Ziyan, et al. "Safe Multi-agent Reinforcement Learning with Natural Language Constraints." arXiv preprint arXiv:2405.20018 (2024).

**Questions:**

Q1: What are the global guarantees on the safety aspect when using the proposed epigraph formulation?

Q2: Do the methods proposed in the empirical evaluation solve the tasks? If so, why the author do not reported the mean global success rate/ mean global return?

Q3: What is the scalability of the proposed formulation? Does the use of GNNs limit the applicability of the proposed methods to a significant number of agents?

---

> ### Author Response · Authors · 2024-11-18
> **Reply (1/3)**
>
> We would like to thank the reviewer for finding our approach novel and for acknowledging our comparison to existing theoretical approaches.
>
> *1. What are the global guarantees on the safety aspect when using the proposed epigraph formulation?*
>
> **A1**: As introduced in Section 4.1 of the original submission, the proposed epigraph formulation is **equivalent** to the original multi-agent constraint optimal control problem.
> Therefore, solving the epigraph formulation has the same global guarantees on the safety aspect as the original problem. For more information on the equivalence, we have added the proof in Appendix G in the revised version.
>
> ---
>
> *2. Do the methods proposed in the empirical evaluation solve the tasks? If so, why the author does not report the mean global success rate/ mean global return?*
>
> **A2**: **We already report the mean global return. The cost $l$ in our paper is equivalent to the negation of the global return**, and is **NOT** the cost in the CMDP setting.
> The cost in our paper, defined in Line 148 of the original paper, refers to the objective function rather than the constraints and is a measurement of **task completion instead of safety**.
> This is the convention used in optimal control (see [α, β, γ]), and can be translated to the reward maximization setting by taking the cost $l$ as the negative of the reward function.
>
> We define the cost function $l$ we use in our experiments using the distance to the goal and the control penalty (Appendix D.2 in the original submission), which correspond to task completion. Hence, lower costs mean that the agents better complete the given task.
>
> The global return = global cumulative cost is **already reported** in Figure 3 and Figure 6. Algorithms that incur less cost (x-axis) solve the task better.
>
> Since our approach is motivated by optimal control techniques and uses an optimal control problem formulation, we maintain our use of the word "cost" to refer to the objective to match the convention of the optimal control community.
>
> However, to prevent confusion, **we have added the following sentence in the revised paper** when defining the cost $l$:
>
> > Given a global cost function $l: \mathcal X\times\mathcal U\to\mathbb R$ describing the task for the agents to accomplish
>
> We have also added a footnote in the revised paper to explain the relationship with the CMDP setting:
>
> > The cost function $l$ is *not* the cost in CMDP. Rather, it corresponds to the negation of the *reward* in CMDP.
>
> ---
>
> *3. This decentralized approach primarily guarantees "constraint satisfaction" rather than task completion, meaning that while the agents may stay safe, they might not achieve the global objective if staying in place satisfies the constraints without addressing the task goals.*
>
> **A3**: As clarified in the above answer, **the cost minimization is concerned with task completion** and is **separate** from satisfying the safety constraints. Theorem 1 states that we can solve for the outer problem — finding the $z$ that optimizes for task performance subject to safety constraints — in a **distributed** way that yields the **same optimal solution** as the original centralized approach and thus achieves the optimal task completion that can be achieved while maintaining safety.
>
> A policy $\pi$ that "stays in place" would be safe and hence have a low constraint value function $V^h$. However, since the task is not being achieved, the cost value function $V^l$ for *task completion* would be high.
> Such a $\pi$ could be optimal if $z$ is large enough, where the _total_ value function
> $$
> V(x, z; \pi) = \max\{ V^h(x, \pi), V^l(x, \pi) - z \} = V^h(x, \pi)
> $$
> would be equal to $V^h$, and hence $\pi$ which optimizes for safety is optimal.
>
> **However**, for a smaller $z$ such that $V(x, z; \pi) = V^l(x, \pi) - z$, this $\pi$ would not be optimal.
> Instead, the optimal policy for this smaller $z$ would need to minimize $V^l$ by completing the task.
> Different values of $z$ then balance between the two objectives.
>
> The _optimal_ $z$, found by solving (10a) in the original version and (13a) in the revised version, or equivalently solved decentralized using Theorem 1, will find the $z$ that results in the lowest cost, i.e., most task completion, while still maintaining safety.
>
> ## References
>
> [α] Todorov, Emanuel. "Compositionality of optimal control laws." NeurIPS 2009.
>
> [β] Chow, Yinlam, et al. "A lyapunov-based approach to safe reinforcement learning." NeurIPS 2018.
>
> [γ] Li, Yingying, Xin Chen, and Na Li. "Online optimal control with linear dynamics and predictions: Algorithms and regret analysis." NeurIPS 2019.

---

> > ### Author Response · Authors · 2024-11-18
> > **Reply (2/3)**
> >
> > *4 (1/2). What is the scalability of the proposed formulation?*
> >
> > **A4 (1/2)**: We have demonstrated the scalability of our formulation in our experiments with 5 and 7 agents (Figure 6).
> > **This matches the number of agents considered in previous MARL tasks** [1, 2, 3].
> > Since our algorithm uses MAPPO as a backbone, the scalability is as good as MAPPO.
> >
> > ---
> >
> > *4 (2/2). Does the use of GNNs limit the applicability of the proposed methods to a significant number of agents?*
> >
> > **A4 (2/2)**: We do not believe GNNs _limit_ the applicability to larger number of agents. On the contrary, there are works [4, 5, 6] that suggest that GNNs **enable** their method to be applied to large numbers of agents.
> >
> > ---
> >
> > *5. On the comparison with existing literature: [A, B, C, D] proposed an in-depth study and benchmarks on the mentioned problem. A brief discussion and an empirical comparison against one or two of these methods are missing.*
> >
> > **A5**: We thank the reviewer for pointing out potentially related works. After a careful reading of the works, we find that:
> >
> > - [A]: This paper presents trust region bounds for optimizing decentralized policies in cooperative Multi-Agent Reinforcement Learning (MARL), and proposes a novel algorithm with a monotonic improvement guarantee based on IPPO and MAPPO. However, **it does not consider safety constraints**, and therefore does not match our problem setting.
> > - [B]: This paper considers the safe MARL under the CMDP setting, and proposes an algorithm that addresses the problem with a "safety budget". However, we did a thorough search online, and to the best of our knowledge, **this paper is not published** anywhere and is also not available on ArXiv. The only version of this paper available online is rejected by ICLR 2023. Therefore, the authors are not able to cite this paper.
> > - [C]: This paper analyzes the duality gap of primal-dual algorithms in MARL theoretically and proposes a primal approach that avoids the duality gap. This paper is relevant to our approach and **we have added it in the related work section of our revised paper**. However, this paper does not contain empirical studies on complex environments similar to our considered settings, so we are unable to compare against this work empirically.
> > - [D]: This paper considers the safe MARL setting but focuses on defining safety constraints based on **natural language descriptions**. The proposed SMALL-MAPPO algorithm uses MAPPO with Lagrange multipliers, which we already compare within the paper.
> > Since the additional natural language contributions of this work added on top of MAPPO with Lagrange multipliers is not relevant to our setting, we choose to not include this work.
> >
> > ## References
> >
> > [1] Gu, Shangding, et al. "Safe multi-agent reinforcement learning for multi-robot control." Artificial Intelligence 319 (2023): 103905.
> >
> > [2] Nayak, Siddharth, et al. "Scalable multi-agent reinforcement learning through intelligent information aggregation." International Conference on Machine Learning. PMLR, 2023.
> >
> > [3] Yu, Chao, et al. "The surprising effectiveness of ppo in cooperative multi-agent games." Advances in Neural Information Processing Systems 35 (2022): 24611-24624.
> >
> > [4] Yu, Chenning, Hongzhan Yu, and Sicun Gao. "Learning control admissibility models with graph neural networks for multi-agent navigation." Conference on robot learning. PMLR, 2023.
> >
> > [5] Tolstaya, Ekaterina, et al. "Multi-robot coverage and exploration using spatial graph neural networks." 2021 IEEE/RSJ International Conference on Intelligent Robots and Systems (IROS). IEEE, 2021.
> >
> > [6] Zhang, Songyuan, et al. "Gcbf+: A neural graph control barrier function framework for distributed safe multi-agent control." arXiv preprint arXiv:2401.14554 (2024).
> >
> > [A] Sun, Mingfei, et al. "Trust region bounds for decentralized ppo under non-stationarity." AAMAS 2023.
> >
> > [B] Wang, Z., Du, Y., Sootla, A., Ammar, H. B., & Wang, J. (2022). CAMA: A New Framework for Safe Multi-Agent Reinforcement Learning Using Constraint Augmentation.
> >
> > [C] Chen, Ziyi, Yi Zhou, and Heng Huang. "On the Duality Gap of Constrained Cooperative Multi-Agent Reinforcement Learning" (2024).
> >
> > [D] Wang, Ziyan, et al. "Safe Multi-agent Reinforcement Learning with Natural Language Constraints." arXiv preprint arXiv:2405.20018 (2024).

---

> > > ### Author Response · Authors · 2024-11-18
> > > **Reply (3/3)**
> > >
> > > *6. Notational consistency in preliminaries*
> > >
> > > **A6**: Thanks for the suggestion! We have changed the optimization variable to be $\pi$ in Section 3.2 to more closely align with our problem setting, and included clarifications that one example of a cost function $J(\pi)$ could be the optimal control problem we introduced earlier.
> > >
> > > ---
> > >
> > > *7. In Eq. (7), there is $k >= \tau$, but is never defined/discussed*
> > >
> > > **A7**: This is a standard notation for defining the value function / cost-to-go for $x_\tau$. The value function for state $x_\tau$ would sum up the costs of states $x_\tau$, $x_{\tau + 1}$,  $x_{\tau + 2}$ and so on, which can be written as
> > > $$\sum_{k \geq \tau} l(x^k, \pi(x^k)) = l(x^\tau, \pi(x^\tau)) + l(x^{\tau+1}, \pi(x^{\tau+1})) + l(x^{\tau+2}, \pi(x^{\tau+2})) + \dots $$
> > >
> > > ---
> > >
> > > *8. Using the term "value function" could be misleading, as we typically denote by "value function" the value of a state.*
> > >
> > > **A8**: Our use of the term "value function" **denotes the value of a state exactly as you have mentioned**.
> > >
> > > Since we consider a different optimization problem than is typically considered in RL, the value of our state cannot be directly computed as
> > > $$
> > > V(x^k) = r^k + V(x^{k+1}),
> > > $$
> > > because our objective function for policy optimization is not the sum of rewards.
> > > Instead, our objective function is (9) in the original = (12) in the revised version.
> > > We can still express the value of a state recursively using the value of the next state, which we have done in Proposition 1.
> > >
> > > To avoid confusion, we have included additional clarifications on this in Section 4.1 in our revised paper.

---

> ### Author Response · Authors · 2024-11-25
> **We are sincerely looking forward to the reply from Reviewer tD17**
>
> Dear Reviewer tD17,
>
> Thank you again for the constructive questions. We have addressed them in our previous replies accordingly. For a brief summary, we have:
>
> 1. Explained that we **did** report the performance of solving the task, which might have been **a very important misunderstanding** of the reviewer.
> 2. Explained the scalability and the use of GNN.
> 3. Discussed about more related works.
> 4. Clarified some notations.
>
> As the discussion period deadline approaches, we sincerely want to know if our reply has addressed your concerns. **If so, would you please kindly raise your score?** We are also happy to answer any further questions you have.

---

> > ### Comment · Reviewer_tD17 · 2024-11-25
> >
> > Thank you very much for your answers that clarify most of my concerns. In my opinion no further interactions are required on the issues raised in my review at this stage.
> >
> > I will carefully consider whether to raise my score considering also the other reviews and the whole discussion.

---

> > > ### Author Response · Authors · 2024-11-25
> > > **Thanks for letting us know that your concerns are addressed!**
> > >
> > > Thank you very much for the reply! We are glad your concerns are addressed.
> > >
> > > We also want to thank the reviewer for carefully reading comments from other reviewers.
> > > We have tried our best to address them as well.
> > >
> > > As a result, **Reviewer $\color{#E24A33}{\textsf{iiYA}}$ has increased their score**.
> > > We have also engaged in constructive discussion with Reviewer $\color{#988ED5}{\textsf{RYsH}}$, which has greatly improved our theoretical presentation.
> > >
> > > We sincerely look forward to your further decision. Thank you again for the valuable feedback.

---

> > > > ### Author Response · Authors · 2024-12-03
> > > > **Kind reminder of the discussion period deadline**
> > > >
> > > > Dear Reviewer tD17,
> > > >
> > > > Thank you again for the valuable questions and the timely reply! We are glad that your concerns are addressed.
> > > >
> > > > In your last reply, you mentioned that you would carefully consider raising your score based on the whole discussion period. As the whole **discussion period ends today** (Dec 2nd at midnight AoE), **would you please kindly let us know your current decision on your score**?
> > > >
> > > > Since your last reply, Reviewer $\color{#E24A33}{\textsf{iiYA}}$ has raised their score and Reviewer $\color{#988ED5}{\textsf{RYsH}}$ has posted additional clarification questions which we have addressed accordingly. We have also updated our paper to make Remark 1 more rigorous. However, we have not received any further replies from Reviewer $\color{#988ED5}{\textsf{RYsH}}$ since our response 8 days ago.
> > > >
> > > > Your opinion matters a lot to us, and we sincerely hope you can kindly raise your score if we have addressed all your concerns.
> > > >
> > > > Best regards,
> > > >
> > > > Authors

---

### Official Review · Reviewer_iiYA · 2024-11-04

**Soundness:** 2
**Presentation:** 3
**Contribution:** 2
**Rating:** 6
**Confidence:** 4

**Summary:**

The paper presents an approach called EFMARL, which extends the epigraph form optimization method to address safe multi-agent reinforcement learning (MARL). Based on the epigraph form, EFMARL targets the problem of achieving zero constraint violations in multi-agent environments by using a centralized training and distributed execution (CTDE) approach. The method is tested in both multi-particle and Safe Multi-agent MuJoCo environments, demonstrating stability and performance in enforcing safety constraints while avoiding performance degradation.

**Strengths:**

1. The paper is well-written and easy to follow.
2. The related work is well investigated.

**Weaknesses:**

1. The Lagrangian multiplier, $\lambda$, may also decrease depending on various Lagrangian version configurations.

2. The performance of the Lagrangian method depends on the parameter settings for different tasks. In my understanding, if we fine-tune the Lagrangian parameters, the Lagrangian method can also perform better.

**Questions:**

1. Can the study provide a convergency analysis?
2. Given the centralized training requirement, what specific computational challenges arise, and how are these managed to ensure scalability with an increasing number of agents?

---

> ### Author Response · Authors · 2024-11-18
> **Reply (1/2)**
>
> We thank the reviewer for their constructive remarks on our paper and for raising important questions.
>
> *1. The Lagrangian multiplier, $\lambda$, may also decrease depending on various Lagrangian version configurations.*
>
> **A1**: This is true **only if the constraint can take negative values**. More specifically, if the CMDP constraint is
> $$
> \sum_{k=0}^\infty c(x^k) \leq d,
> $$
> then the Lagrange multiplier update is $\sum_{k=0}^\infty c(x^k) - d$, which could be negative depending on the choice of $c$ and $d$.
>
> **However**, in the **zero** constraint violation setting that we are targeting, $c(x^k) \geq 0$ and $d = 0$. In this setting, $\lambda$ does not decrease because $\sum_{k=0}^\infty c(x^k) - d \geq 0$
>
> In the revised paper, we have expanded on this explanation (see Paragraph "Comparison with the Lagrangian method" in Section 3.2).
>
> ---
>
> *2. The performance of the Lagrangian method depends on the parameter settings for different tasks. In my understanding, if we fine-tune the Lagrangian parameters, the Lagrangian method can also perform better.*
>
> **A2**: We agree with the reviewer on this observation. This also matches what we have observed in our experiments — for the Lagrangian method, the performance using the same hyperparameters varies greatly on different tasks.
> This is an **advantage** of our algorithm since we only need to use one set of hyperparameters to consistently outperform the baselines in all environments.
>
> In our experiments, we have explored the following settings:
>
> 1. **The Lagrange multiplier is held static during training.** In this case, larger values of Lagrange multipliers result in conservative policies that have a high cost (InforMARL-L(5)), while small values of Lagrange multipliers result in aggressive policies that violate safety constraints (InforMARL-L(1)). Our method, however, consistently outperforms both of them without hyperparameter tuning. In addition, using a non-optimal Lagrange multiplier can cause the optimal solution to **change** (Section 5.2, paragraph 2). Therefore, even if the optimal policy is found, this is **not** the optimal policy of the original problem.
> The optimal policy of the original problem is recovered only if the Lagrange multiplier used is optimal.
> However, searching for the optimal Lagrange multiplier can require many runs, which we would like to avoid, especially in tasks where each run takes 6 hours to run.
>
> 2. **The Lagrange multiplier is learned during training.** We have shown in Figure 5 that this causes unstable training. This empirical result is also supported by our theoretical analysis in the paragraph "Comparison with the Lagrangian method" in Section 3.2.
>
> Our proposed method outperforms both settings. Therefore, as claimed in the contributions, our algorithm achieves stable training and does not require hyperparameter tuning.
>
> ---
>
> *3. Can the study provide a convergence analysis?*
>
> **A3**: Thanks for the suggestion! We have added a convergence analysis in Appendix F to our revised version that shows that the policy of the inner RL problem (13b in the revised version) converges almost surely to a locally optimal policy.
>
> In short, since the training of our algorithm is centralized, we can reduce the inner problem to a single-agent avoid RL problem by defining a suitable augmented state and a set of augmented dynamics. We then obtain the result by following the convergence analysis of single-agent avoid RL from existing works.

---

> > ### Author Response · Authors · 2024-11-18
> > **Reply (2/2)**
> >
> > *4. Given the centralized training requirement, what specific computational challenges arise, and how are these managed to ensure scalability with an increasing number of agents?*
> >
> > **A4**: The alternative to centralized training would be decentralized training.
> > Decentralized training has benefits in terms of the **communication** requirements since each agent can train their own policy without needing to communicate with other agents.
> > However, from a **computational** standpoint, both centralized and decentralized methods can take advantage of multiple computational nodes, so there are no centralized-training specific computational challenges.
> >
> > On the other hand, decentralized training can actually yield **worse computational complexity**, since each agent will have their own (decentralized) value function compared to just a single value function since the parameters are no longer shared.
> > Not only does the memory required for storing the value function parameters now scale with the number of agents (as opposed to being constant), but also the memory required for computing the forward and backward pass for the value function updates will also now scale with the number of agents.
> >
> > Finally, since works in the literature have shown that centralized training methods perform better than their decentralized counterparts [1], we have chosen not to use a decentralized training method for our approach.
> >
> > ## References
> >
> > [1] Yu, Chao, et al. "The surprising effectiveness of ppo in cooperative multi-agent games." Advances in Neural Information Processing Systems 35 (2022): 24611-24624.

---

> > > ### Comment · Reviewer_iiYA · 2024-11-24
> > >
> > > Thank you for the authors' response, which largely addressed my concerns. As a result, I have decided to raise my score to 6. However, for better clarity and presentation, it is important to clearly define the notation used in the manuscript. For instance, whether $J$ represents the objective or the cost, and $h$ represents the constraints, should be explicitly clarified. Additionally, the relationship between these terms and the reward should be presented more clearly, as they are occasionally used interchangeably in the manuscript.

---

> > > > ### Author Response · Authors · 2024-11-24
> > > > **Author Reply**
> > > >
> > > > Thank you for raising your score! We are glad we have mostly addressed your concerns.
> > > >
> > > > **Would you please kindly let us know what changes you would like to see to further raise your score?**
> > > >
> > > > ---
> > > >
> > > > **Q1**: Clarify whether $J$ represents the objective or the cost, and $h$ represents the constraints.
> > > >
> > > > **R1**: Assuming this is referring to Section 3.2 where we explain the epigraph form (since this is the only place where we have used $J$ in the entire manuscript), **$J$ is simply the objective of (3).** We can take the objective to be the cost function (Line 178), so **the objective _is_ the cost. There is no difference between the two.**
> > > >
> > > > In Section 3.2, we also state very clearly that $h$ represents the constraints (Line 179).
> > > >
> > > > > Given a constrained optimization problem with **objective function $J$** (e.g., $J = \sum_{k=0}^\infty l$ as in (2a)) and **constraints $h$** (e.g., (2b))
> > > >
> > > > **If the reviewer believes this sentence can be made clearer, please let us know.** Thank you!
> > > >
> > > > ---
> > > >
> > > > **Q2**: The relationship between these terms and the reward should be presented more clearly, as they are occasionally used interchangeably in the manuscript.
> > > >
> > > > **R2**: **We have not used the word "reward" in the main pages of the manuscript** outside of the related works,
> > > > except to explain in a footnote that the cost we consider here is the negative of the reward in the CMDP setting.
> > > >
> > > > **There is no concept of reward in our work**.
> > > >
> > > > The word was used once in Appendix E.2.2 to match the original description of the safe multi-agent MuJoCo environment, but we have clarified that we take the cost to be the negative of the reward in the revised manuscript.

---

### Author Response · Authors · 2024-11-18
**Response to all**

We thank the reviewers for their valuable comments.
We are excited that the reviewers identified the novelty of our technical contributions (**all** reviewers) on an important problem ($\color{#348ABD}{\textsf{tD17}}$), acknowledged our extensive experiments ($\color{#348ABD}{\textsf{tD17}}$, $\color{#988ED5}{\textsf{RYsH}}$), and found the paper well-written with good presentation ($\color{#E24A33}{\textsf{iiYA}}$, $\color{#348ABD}{\textsf{tD17}}$).
We believe EFMARL takes a significant step towards hyperparameter-insensitive algorithms for solving safe multi-agent reinforcement learning problems in the _zero_ constraint violation setting.

---

## 1. More Concise Proof

As _all_ reviewers have recognized our technical novelty, the primary criticism stems the proof of Theorem 1 being difficult to follow ($\color{#988ED5}{\textsf{RYsH}}$).
In the revision we **revise our proof of Theorem 1 in Appendix B to make it much more concise**.

## 2. New Experiments

We also perform a new experiment on the effect of the sampling interval of $z$ in Appendix E.7:
- Even if the estimate of the maximum cost $z_\mathrm{max}$ is off by $50\\%$, the safety and cost does not change much.
- If we are even more conservative by doubling its value, we remain safe but the cost becomes slightly worse.
- If we severely underestimate the maximum cost, safety degrades.

---

We hope that the new presentation is clearer in presenting the contributions of our method in improving both training stability and performance for safe multi-agent problems in the zero constraint violation setting.
We have tried our best to resolve all raised questions in the individual responses below. If you have any additional questions/comments/concerns, please let us know. We appreciate the reviewer's precious time in providing their valuable feedback.

---

### Meta-Review · Area_Chair_T3xu · 2024-12-19

**Metareview:**

Distributed Epigraph Form Multi-Agent Safe Reinforcement Learning

Summary: This paper proposes a new method for tackling the multi-agent constrained reinforcement learning problem. The paper highlights challenges in existing methods, such as Lagrangian-based approaches, that struggle with training instability when the constraint violation threshold is set to zero. The proposed method, EFMARL, addresses these issues by extending the epigraph form technique—commonly used in single-agent settings—to the multi-agent setting, specifically for decentralized execution under centralized training. The algorithm decomposes the problem into an inner optimization problem, solved during training, and an outer problem, solved during execution, ensuring stable training while adhering to safety constraints. The method is robust to environmental changes, requiring a constant set of hyperparameters across various tasks.  Experiments on benchmark environments, including MuJoCo setting, illustrate the performance of EFMARL compared to existing approaches.

Comment: We have received 3 expert reviews, with the scores 3, 5, 6, and the average score is 4.67.

The reviewers have given positive comments about multiple aspects of the paper. This includes the epigraph-based formulation of safe learning and the use of graph neural network (GNN) architecture for algorithm implementation. However, based on the reviewers' comments, the weaknesses outweigh these positive aspects.

Reviewer tD17 has pointed out multiple missing references that are relevant to this work and remarked on weaknesses in the empirical evaluation, especially with respect to these works. Reviewer RYsH has pointed out multiple weaknesses in the technical analysis including lack of rigor and presence of errors/ambiguities in Proposition 1 and Theorem 1. Reviewer tD17 has expressed concern about the lack of a thorough empirical evaluation that addresses both task completion and scalability challenges. The reviewers are also not completely satisfied with the presentation of the paper and have commented that the details are often difficult to follow. In summary, while the paper introduces an interesting and potentially impactful approach, significant weaknesses in theoretical rigor, empirical validation, and clarity reduce its contribution. I recommend the authors address these issues in a resubmission.

**Additional Comments On Reviewer Discussion:**

Please see the "Comments" in the meta-review.

---

### Decision · Program_Chairs · 2025-01-22

Reject